# ATOM: A Pretrained Neural Operator for Multitask Molecular Dynamics

**Luke Thompson**
The University of Sydney

**Davy Guan**
Data61 CSIRO

**Dai Shi**
University of Cambridge

**Slade Matthews**[*]
The University of Sydney

**Junbin Gao**[*]
The University of Sydney

**Andi Han**[*][†]
The University of Sydney

## Abstract

Molecular dynamics (MD) simulations underpin modern computational drug discovery, materials science, and biochemistry. Recent machine learning models provide high-fidelity MD predictions without the need to repeatedly solve quantum mechanical forces, enabling significant speedups over conventional pipelines. Yet many such methods typically enforce strict equivariance and rely on sequential rollouts, thus limiting their flexibility and simulation efficiency. They are also commonly single-task, trained on individual molecules and fixed timeframes, which restricts generalization to unseen compounds and extended timesteps. To address these issues, we propose Atomistic Transformer Operator for Molecules (ATOM), a pretrained transformer neural operator for multitask molecular dynamics. ATOM adopts a quasi-equivariant design that requires no explicit molecular graph and employs a temporal attention mechanism, enabling accurate parallel decoding of multiple future states. To support operator pretraining across chemicals and timescales, we curate TG80, a large, diverse, and numerically stable MD dataset with over 2.5 million femtoseconds of trajectories across 80 compounds. ATOM achieves state-of-the-art performance on established single-task benchmarks, such as MD17, RMD17 and MD22. After multitask pretraining on TG80, ATOM shows exceptional zero-shot generalization to unseen molecules across varying time horizons. We believe ATOM represents a significant step toward accurate, efficient, and transferable molecular dynamics models.

## 1 Introduction

Molecular dynamics (MD) serves as a computational microscope of atomic motion and is now integral to drug discovery and materials science pipelines (Dror et al., 2012; De Vivo et al., 2016). In ab initio molecular dynamics, quantum-mechanical density functional theory (DFT) is used to compute atomic forces, and the resulting equations of motion are integrated to generate high-fidelity trajectories. However, DFT's computational complexity scales at least cubically with the number of atoms, and relies on double-precision arithmetic that limits GPU acceleration (Kresse & Furthmüller, 1996; Stein et al., 2020; Li et al., 2024).

Neural approaches have recently emerged as a promising solution to the scalability bottleneck. *Equivariant* architectures, in particular, encode physical symmetries to model interatomic dynamics, achieving ab initio-level accuracy at significantly reduced computational cost (Batzner et al., 2022; Musaelian et al., 2022; Batatia et al., 2022; 2023; Xu et al., 2024). While equivariance is often deemed essential for improving generalization, strict symmetry preservation involves substantial tradeoffs (Xu et al., 2024; Schreiner et al., 2023). Architectures that enforce exact equivariance at every layer often increase computational overhead, restrict model expressivity, and complicate optimization (Fuchs et al., 2020; Brehmer et al., 2023; Elhag et al., 2025). It is unclear whether symmetry constraints can be relaxed without sacrificing accuracy for molecular dynamics.

---

[*]These authors contributed equally as senior authors.
[†]Correspondence to: `andi.han@sydney.edu.au`

Furthermore, most existing methods for molecular dynamics are *autoregressive*, predicting the next state based on the current one (Köhler et al., 2019; Fuchs et al., 2020; Thiemann et al., 2025). Autoregressive approaches often struggle to capture long-horizon temporal dependencies and accumulate error as the prediction horizon grows (Bengio et al., 2015; Bergsma et al., 2023; Taieb & Atiya, 2016). Inference speeds are also constrained by the need for sequential integration, failing to exploit modern, highly parallel compute architectures. One exception is Equivariant Graph Neural Operator (EGNO) (Xu et al., 2024), which models the entire trajectory with neural operator learning. Nevertheless, EGNO enforces strict equivariance and is single-task in nature, i.e., it is trained and evaluated on trajectories of each molecule separately with a fixed time horizon, which limits *zero-shot generalization* to unseen molecules or timeframes.

**Our Main Contributions.** In this work, we address the above issues regarding equivariance, autoregression, and zero-shot generalization within a unified framework, which we call *Atomistic Transformer Operator for Molecules (ATOM)*. To this end, we propose a pre-trained neural operator with a transformer backbone for molecular dynamics and introduce a new MD dataset, TG80, which is both chemically diverse and numerically stable for multitask pretraining and benchmarking.

- *Design innovations.* ATOM is *quasi-equivariant* by employing an equivariant lifting layer that produces symmetry-aware features, while allowing subsequent transformer blocks to be unconstrained for flexibility and expressiveness. Unlike autoregressive models, ATOM allows *parallel decoding* of molecule states across multiple timesteps, directly learning the trajectory operator. By encoding time lags via a novel temporal rotary position embedding, ATOM enhances temporal interpolation and extrapolation, enabling robust predictions across multiple time horizons. Finally, ATOM requires no predefined molecular graph and operates directly on *point clouds*, naturally accommodating long-range spatial interactions without the need for hand-crafted connectivity.

- *Performance highlights.* ATOM sets new state-of-the-art on single-task MD benchmarks. For larger, sparsely connected molecules in MD22, ATOM significantly outperforms existing graph-based baselines by capturing the long-range atomic interactions. In the multitask regime, we pretrain ATOM on TG80 trajectories from multiple molecules and varying timeframes, demonstrating significant zero-shot transfer to both unseen molecules and timesteps, improving existing baselines by 39.75% on average. This achieves performance on par with existing specialized baselines tailored for such molecules and timeframes. To the best of our knowledge, this is the first method that demonstrates such generalization capability in molecular dynamics.

We believe our work represents a shift in molecular dynamics modeling, where we demonstrate the potential of quasi-equivariance designs and zero-shot generalization to out-of-domain systems, which is enabled by the comprehensive TG80 MD dataset. ATOM and TG80 are available at this repository.

## 2 RELATED WORK

**Equivariant Neural Networks.** Equivariance (to transformations such as rotation, reflection, and translation) has emerged as an essential physics-informed prior for deep learning models on molecular data (Bronstein et al., 2021; Duval et al., 2023). Early works employed convolutional approaches to achieve translation equivariance in E(3) (Weiler et al., 2018; Wu et al., 2020) or tensor product attention and spherical harmonics to enforce roto-translational equivariance in SE(3) (Fuchs et al., 2020; Thomas et al., 2018). In contrast, message passing neural network (MPNN) frameworks, such as Equivariant Graph Neural Network (EGNN) and others (Garcia Satorras et al., 2021; Gasteiger et al., 2021; Huang et al., 2022), achieve equivariance by operating on strictly equivariant features, such as inter-node distances and directions. While effective, MPNNs typically assume a fixed molecular graph. This is problematic when the underlying structure contains non-local interactions and dynamic bonding effects (e.g., resonances, transient interactions), which render predefined graphs inaccurate over time (Knutson et al., 2022; Luo et al., 2021). To address this issue, we model molecules as point clouds, with our attention represented as a fully connected graph that allows unrestricted information propagation across the molecule.

**Time-coarsened Molecular Dynamics** Time coarsening is a coarse-graining method which preserves molecular structure, but compresses many short integration steps into a few large-stride updates to reduce the cost of long-time simulation (Kmiecik et al., 2016). Stochastic coarse-graining approaches often learn transition kernels on configuration space, bypassing explicit integration of the equations

of motion. Klein et al. (2023) learns such a kernel with a normalizing flow and uses it as an MCMC proposal targeting the Boltzmann distribution, Hsu et al. (2024) uses a conditional diffusion model to learn a transition probability matrix, and Yu et al. (2025) uses flow-matching to learn a vector field transporting current states to future states. Closer to our framework, deterministic methods such as MDNet (Zheng et al., 2021) and TrajCast (Thiemann et al., 2025) learn a GNN and EGNN, respectively, which autoregressively predict fixed strides 10-100 times larger than those of MD integrators. Bigi et al. (2025) incorporates Hamiltonian structure and explicit energy-conservation. Most of the methods require direct force learning and are sequential in nature, while ATOM may be interpreted as a *force-free* deterministic coarse-graining approach, wherein temporal pushforward is approximated by a learned propagation operator which is decoded *in parallel*.

**Neural Operators.** Neural operators are deep learning methods for learning operators between function spaces (Kovachki et al., 2021). A wide variety of architectures have been proposed for such operator learning. Notably, Fourier Neural Operator (FNO) (Li et al., 2021) learns an operator in the Fourier domain, while its derivatives G-FNO (Helwig et al., 2023) and PINO (Li et al., 2023b), respectively, add group equivariance and physics-informed properties. Xu et al. (2024) bridges this framework with molecular dynamics by recasting the task as learning a propagation operator that evolves historical atomic positions into their future configurations. Specifically, EGNO (Xu et al., 2024) integrates EGNN and FNO layers to learn dynamic trajectories, capturing spatial and temporal correlations. Recently, transformer neural operators (Bryutkin et al., 2024; Hao et al., 2023; Li et al., 2023a) have surpassed the performance of FNO in most partial differential equation (PDE) tasks. Notably, OFormer (Li et al., 2023a) uses a linear Galerkin-type attention mechanism, which omits the softmax and instead interprets the latent column vectors as basis functions. General Neural Operator Transformer (GNOT) (Hao et al., 2023) employs a novel subquadratic cross-attention methodology to integrate multiple feature types (e.g., shape and point relationships) into their transformer blocks. With ATOM, we unify the MD problem formulation and temporal discretization approach introduced by EGNO with the increased representational power of transformers in operator settings.

**MD Benchmarks.** Research on graph machine learning for molecular dynamics suffers from poor benchmarking (Bechler-Speicher et al., 2025). For example, despite the fact that MD17 Benzene exhibits non-physical noise approximately 1000 times higher compared to other compounds (Christensen & von Lilienfeld, 2020), it is still regularly employed to benchmark new models (Bihani et al., 2023; Huang et al., 2022; Liao & Smidt, 2023; Xu et al., 2024). The practical relevance of single-task learning on these datasets is also dubious, as predicting trajectories for molecules with existing numerical solutions offers minimal benefit. We believe the strengths of neural approaches emerge in transfer learning, where models generalize to unseen compounds, thereby circumventing the computational costs associated with explicit numerical simulations. This motivates our development of TG80 to facilitate multitask dynamics learning across molecular systems.

## 3 ATOMISTIC TRANSFORMER OPERATOR FOR MOLECULES (ATOM)

In this section, we first introduce the problem formulation (Section 3.1) and then propose the framework of ATOM by introducing the key model and training designs (Section 3.2). We then discuss the multitask pretraining for ATOM and introduce TG80 MD dataset (Section 3.3).

### 3.1 PROBLEM FORMULATION

We follow (Xu et al., 2024) to cast molecular dynamics prediction as operator learning. We model a molecule of $N$ atoms as a point cloud in $\mathbb{R}^3$, which we denote as $\mathcal{G}^{(t)}$ for a given system state time $t$. In particular, we write $\mathcal{G}^{(t)} = (\mathbf{x}_i^{(t)}, \mathbf{v}_i^{(t)})_{i=1}^N$ that represent molecules in terms of the atom positions $\mathbf{x}$ and velocities $\mathbf{v}$. Our objective is to predict a future trajectory $\mathcal{G}^{(t+\Delta t)}$, where $\Delta t \in [0, \Delta T]$.

Similar to (Xu et al., 2024), we focus on predicting the position states only. Let $\mathcal{U} \colon [0, \Delta T] \to \mathbb{R}^{N \times 3}$ be the trajectory function mapping $\Delta t$ to $U(\Delta t) \in \mathbb{R}^{N \times 3}$ representing molecule positions $\Delta t$ in the future. We assume a solution operator $F^\dagger \colon \mathcal{G}^{(t)} \to \mathcal{U}$ exists which provides the underlying future trajectory given system states at $t$. Thus, the goal of molecular dynamics prediction becomes training a neural operator $F_\theta(\mathcal{G}^{(t)})$ to approximate the target trajectory function $F^\dagger(\mathcal{G}^{(t)})$: $\min_\theta \mathbb{E}_{\mathcal{G}^{(t)}} \mathcal{L}\big(F_\theta(\mathcal{G}^{(t)})(t), F^\dagger(\mathcal{G}^{(t)})(t)\big)$, for some loss function $\mathcal{L} \colon \mathcal{U} \times \mathcal{U} \to \mathbb{R}$. Here, expectation is with respect to the different initial states. By discretizing over the temporal domain and considering

$L_2$ loss, we optimize the neural operator with a discretized temporal sampling of the states:

$$\min_{\theta} \frac{1}{P} \sum_{p=1}^{P} \mathbb{E}_{\mathcal{G}^{(t)}} \left\| F_{\theta}\left(\mathcal{G}^{(t)}\right)(\Delta t_p) - F^{\dagger}\left(\mathcal{G}^{(t)}\right)(\Delta t_p) \right\|_2^2. \quad (1)$$

where $\{\Delta t_1, ..., \Delta t_p\}$ are discrete timesteps. We replace the true future state $F^{\dagger}(\mathcal{G}^{(t)})(\Delta t_p)$ with the known future ground truth node positions $\mathbf{x}^{(t+\Delta t_p)}$ for $\Delta t_p \in [0, \Delta T]$.

**Quasi-equivariance.** We formally define quasi-equivariance, motivated by (Elhag et al., 2025).

**Definition 3.1** ($\varepsilon$-quasi-equivariance). *We call a function $f : \mathcal{X} \to \mathcal{Y}$, $\varepsilon$-quasi-equivariant with respect to group $G$ if it satisfies $\mathbb{E}_{x \in \mathcal{X}} \| \int_G f(\phi(g)(x)) d\mu(g) - \int_G \rho(g)(f(x)) d\mu(g) \| \leq \varepsilon$, where $\mu$ denotes the normalized Haar measure.*

In practice, we approximate the group integration with Monte Carlo samples from $G$.

**Single- and multitask.** Unlike prior works (Schreiner et al., 2023; Xu et al., 2024), we consider both single-task and multitask settings. *Single-task* refers to the case where a separate model is independently trained and evaluated on each molecule and fixed timeframes. This corresponds to the conventional practice in molecular dynamics benchmarks. *Multitask* instead pretrains one unified model on several molecules across varying time lags and evaluates out-of-domain trajectories on unseen molecules, thereby directly testing zero-shot cross-molecule generalization. Under a multitask setting, the objective (1) computes the expectation over trajectories of multiple molecules.

### 3.2 ATOM MODEL AND TRAINING DESIGN

Here we outline the pipeline of ATOM. At its core is an *equivariant lifting* layer (Section 3.2.1), which maps atomic positions, velocities and their phase features into a richer embedding space while preserving symmetry under the Euclidean group $E(3)$. The lifted embeddings are then processed by the ATOM attention block, which applies *heterogeneous attention* over positions, velocities, and phase features with chemical augmentation (Section 3.2.2). To capture temporal dynamics, we incorporate a *temporal rotary position embedding* (T-RoPE) (Section 3.2.2) that depends only on time lags and is shared across atoms, ensuring translation invariance in time and permutation invariance within each molecule. The parameterized ATOM can be written by

$$F_{\theta} := \mathcal{P} \circ \sigma(\mathcal{K}_L) \circ \cdots \circ \sigma(\mathcal{K}_1) \circ \mathcal{Q}$$

where $\mathcal{Q}, \mathcal{P}$ denotes the equivariant lifting and projection operators respectively. $\mathcal{K}_l, l = 1, ..., L$ are the data-dependent kernels induced by cross attention (See Appendix G.1), and $\sigma$ denotes some nonlinear activation function.

Finally, to counter numerical noise in training trajectories, we inject randomly sampled position and velocity perturbations during training (Section 3.2.3), which improves robustness and acts as a regularizer against overfitting. The overall pipeline of ATOM is in Figure 1.

### 3.2.1 $E(3)$ EQUIVARIANT LIFTING

To model atomic states in a symmetry-respecting way, each atom is encoded with its 3D position and velocity, augmented with their norms: $\mathbf{x} = (x, y, z, \sqrt{x^2 + y^2 + z^2})$, $\mathbf{v} = (v_x, v_y, v_z, \sqrt{v_x^2 + v_y^2 + v_z^2})$. To construct higher-dimensional features that remain consistent with $E(3)$ symmetry, we apply *equivariant lifting* that maps the inputs through learnable functions that preserve group actions. Specifically, we use $E(3)$-equivariant linear layers (Geiger & Smidt, 2022) that lift the position and velocity vectors to a feature space wherein they satisfy equivariance constraints by construction. We further construct phase space featrues for each atom by augmenting the position and velocity vectors with atomic number, which is subsequently processed by a learnable equivariant layer to obtain a lifted representation. The final lifted embedding for a molecule is given by $(\mathbf{X}, \mathbf{V}, \mathbf{Z}) \in \mathbb{R}^{3 \times NP \times d_v}$ corresponding to position, velocity and phase features. The second dimension aggregates nodes and time for attention and $d_v$ is the embedding space dimension.

We highlight that after the equivariant lifting layer, we do not enforce equivariance in the subsequent transformer blocks. This relaxation improves performance over fully equivariant designs while still showing robustness to random trajectory rotations versus non-equivariant baselines (see Section 4.4).

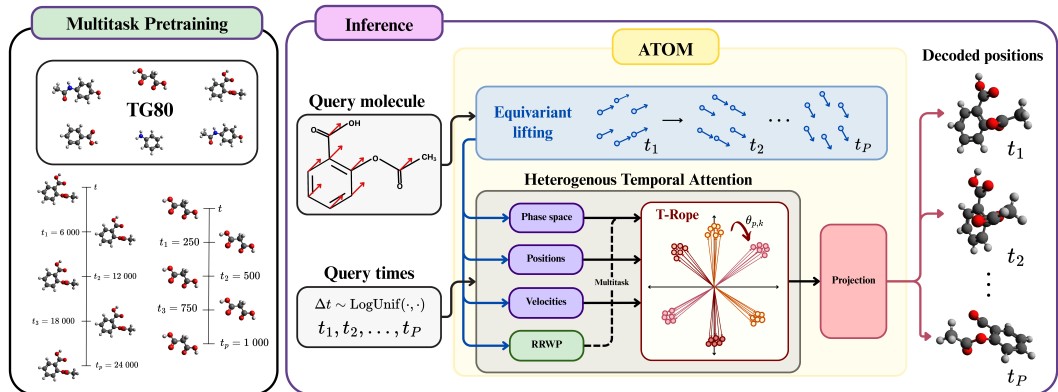

Figure 1: **ATOM Pipeline**. We pretrain ATOM on the TG80 dataset across multiple molecules with stochastic time lags. At inference, ATOM takes a query molecule and timestamps and directly outputs corresponding molecular states.

### 3.2.2 ATOM HETEROGENEOUS TEMPORAL ATTENTION

We employ a heterogeneous temporal attention mechanism to enable mixing between multiple features $(\mathbf{X}, \mathbf{V}, \mathbf{Z}) \in \mathbb{R}^{3 \times NP \times d_v}$ across spatial and temporal dimensions. We use the phase space embedding $\mathbf{Z}$ as the query and attend to the key-value pairs formed from all features $\mathbf{X}, \mathbf{V}, \mathbf{Z} \in \mathbb{R}^{NP \times d_v}$. In Figure 5, we show that this improves performance by 6.36% over standard self-attention for single-task prediction. To encode temporal information, we introduce Temporal RoPE (T-RoPE), adapting Rotary Position Embedding (RoPE) (Su et al., 2023) to irregular time lags by scaling the phase rotations according to cumulative timestamps constructed from per-step increments $\Delta t$.

Let the hidden dimension per head be $d_h$ (even). We define frequencies $\omega_k = b^{-2k/d_h}$ for $k = 0, \ldots, d_h/2 - 1$. Given per-step time increments $\{\Delta t_p\}_{p=1}^P$, we build timestamps $t_p = t + \sum_{r=1}^p \Delta t_r$, and assign a *single* rotation to all $N$ atoms at timestep $p$: $\mathbf{R}_p = \mathrm{diag}\left(\mathbf{R}(\theta_{p,0}), \ldots, \mathbf{R}(\theta_{p,d_h/2-1})\right) \in \mathbb{R}^{d_h \times d_h}$, where $\theta_{p,k} = \frac{\omega_k}{\tau}\left(t_p - t_0\right)$ and $\mathbf{R}(\theta) \in \mathbb{R}^{2 \times 2}$ is the rotation matrix with angle $\theta$ and $\tau > 0$ is a timescale hyperparameter. Suppose the query molecule state at time $p$ is given as $\mathbf{Q}_p \in \mathbb{R}^{N \times d_h}$ and key molecule state at time $p'$ is $\mathbf{K}_{p'} \in \mathbb{R}^{N \times d_h}$. We apply $\mathbf{R}_p, \mathbf{R}_{p'}$ to $\mathbf{Q}_p, \mathbf{K}_{p'}$ respectively so that the rotary dot product $\mathbf{Q}_p \mathbf{R}_p (\mathbf{K}_{p'} \mathbf{R}_{p'})^\top$ depends only on the time interval $t_{p'} - t_p$. This makes attention *translation invariant* in time, which allows for interpolation and extrapolation across irregular increments $\{\Delta t_p\}$. In addition, sharing the same $\mathbf{R}_p$ across all $N$ atoms in a molecule ensures *permutation-invariance* within a timestep. For aggregated query and key matrices $\mathbf{Q}, \mathbf{K} \in \mathbb{R}^{NP \times d_h}$, we denote the application of temporal RoPE across $P$ timesteps and $N$ atoms as T-RoPE($\mathbf{Q}$), T-RoPE($\mathbf{K}$) $\in \mathbb{R}^{NP \times d_h}$.

Specifically, a single-head attention layer of ATOM computes

$$\sum_{\mathbf{F} \in \{\mathbf{X}, \mathbf{V}, \mathbf{Z}\}} \gamma_{\mathbf{F}} \, \mathrm{softmax}\left(\frac{\mathrm{T\text{-}RoPE}(Q(\mathbf{Z})) \, \mathrm{T\text{-}RoPE}(K(\mathbf{F}))^\top}{\sqrt{d_h}}\right) V(\mathbf{F}),$$

where $Q(\cdot), K(\cdot), V(\cdot)$ represent the query, key and value projections. We introduce learnable weights $\gamma_{\mathbf{F}}$ to modulate the relative importance of each feature. In Appendix G.1, we show that heterogeneous attention is equivalent to a kernel integral operator and discuss its properties.

### 3.2.3 TRAINING WITH LABEL NOISE REGULARIZATION

Many DFT datasets are inherently noisy (Christensen & von Lilienfeld, 2020), and MD models can overfit to this noise. Motivated by the regularization effect of label noise (Damian et al., 2021; HaoChen et al., 2020), we augment the observed node positions $\mathbf{x}$ and velocities $\mathbf{v}$ by random Gaussian noise $\boldsymbol{\xi}_{\mathbf{x}}, \boldsymbol{\xi}_{\mathbf{v}} \sim \mathcal{N}(\mathbf{0}, \sigma^2 \mathbf{I})$ during training. Let $\mathcal{G}_{\boldsymbol{\xi}}^{(t)} = (\mathbf{x}_i^{(t)} + \boldsymbol{\xi}_{x,i}, \mathbf{v}^{(t)} + \boldsymbol{\xi}_{v,i})$ be the

noised initial state at time $t$. We minimize the following regularized loss

$$\min_\theta \frac{1}{P} \sum_{p=1}^{P} \mathbb{E}_{\mathcal{G}^{(t)}, \boldsymbol{\xi}, \boldsymbol{\xi}_x^p} \left\| F_\theta \left( \mathcal{G}_{\boldsymbol{\xi}}^{(t)} \right) (\Delta t_p) - (\mathbf{x}^{(t+\Delta t_p)} + \boldsymbol{\xi}_x^p) \right\|_2^2 .$$

A similar strategy has also appeared in graph neural network (GNN)-based MD models and neural operator pretraining (Dauparas et al., 2022; Zhou et al., 2024a; Hao et al., 2024). We only apply noise augmentation during training and evaluate on the unperturbed ground-truth trajectories.

**Comparison to EGNO**. We highlight that ATOM adopts different design choices compared to EGNO. First, EGNO is an EGNN operating on fixed bond connectivity, whereas ATOM uses an E(3)-equivariant lifting layer followed by globally connected point-cloud attention, which better handles long-range and sparsely bonded interactions. Second, EGNO is strictly equivariant end-to-end, while ATOM is quasi-equivariant, enforcing equivariance only in the lifting stage and relaxing it in deeper transformer layers, which our ablations show improves accuracy. Third, EGNO models time via Fourier temporal convolution, whereas ATOM uses Temporal RoPE, allowing translation-invariant handling of irregular time gaps and stronger temporal extrapolation. T-RoPE also uniquely allows modifying the time-horizon $\Delta T$ *at inference* by modulating the rotary phases (Section 3.2.2). Consequently, a pretrained ATOM can be evaluated at arbitrary $\Delta T$ values without retraining.

### 3.3 MULTITASK ATOM PRETRAINING AND TG80 DATASET

This section adapts ATOM for the multitask setting, where the aim is to predict future trajectories for unseen molecules. In order to more effectively distinguish molecules, we construct a radius graph of 1.6 Å based on atomic positions, and apply *random walk positional encoding* (Ma et al., 2023; Lobato et al., 2021) to augment the phase vector $\mathbf{z}$. We describe the process in detail in Appendix D.2 and highlight that such a graph depends only on atomic positions, not chemical bonds.

During multitask training, each mini-batch contains trajectories from multiple molecules. In addition, we perform random sampling for the time lags $\Delta t$ from a log-uniform distribution between $\Delta t_{\min}$ and $\Delta T$, namely $\Delta t \sim \mathrm{LogUnif}(\Delta t_{\min}, \Delta T)$. This aims to enhance the robustness of interpolation and extrapolation in the temporal domain, a consideration that has been similarly explored in (Schreiner et al., 2023). Let $\mathcal{M}$ denote the set of training molecules and let $\mathcal{G}_m^{(t)}$ represent the state of molecule $m \in \mathcal{M}$ at timestamp $t$. We can write the pretraining multitask objective as

$$\min_\theta \frac{1}{|\mathcal{M}|} \sum_{m \in \mathcal{M}} \mathbb{E}_{\mathcal{G}_m^{(t)}, \, \Delta t \sim \mathrm{LogUnif}(\Delta t_{\min}, \, \Delta T)} \left\| F_\theta \left( \mathcal{G}_m^{(t)}, \Delta t \right) (\Delta t) - \mathbf{x}_m^{(t+\Delta t)} \right\|_2^2 ,$$

where we take expectation with respect to initial states of multiple molecules in the training set, as well as the time lags. Here, ATOM also takes a time lag, $\Delta t$, as input, to modulate T-RoPE phase.

**TG80 Dataset.** To facilitate pretraining of our neural operator, we introduce **TG80**, a superset of the MD17 dataset. The initial seed set comprises 40 molecules: 8 MD17 compounds and 32 additional drug-like molecules selected through expert review. We then augment the seed molecules with structurally similar molecules from the PubChem dataset of 173 million compounds (Bolton et al., 2011). Accepted candidates had an ECFP-4 Tanimoto similarity between 0.875 and 0.925 to at least one seed molecule, and no more than 0.80 similarity to previously accepted molecules, alongside other criteria detailed in Appendix C.4 (Landrum et al., 2025; Rogers & Hahn, 2010; Rogers & Tanimoto, 1960). These thresholds follow common practice in the literature, balancing diversity while avoiding collapse into overly narrow chemical subspaces (Matter, 1997; Menke et al., 2021; Eastman et al., 2023; Harper et al., 2004; Zhang et al., 2023).

We generate all trajectories using ORCA V6.01 (Neese, 2022) with the PBE functional (Perdew et al., 1996), def2-SVP basis set (Weigend & Ahlrichs, 2005), Δ4 dispersion corrections (Caldeweyher et al., 2019; 2020; Wittmann et al., 2024) at one femtosecond resolution, 300K temperature, in vacuum. This resembles an enhanced RMD17, with more modern dispersion corrections to improve stability and allow for a larger step size (Christensen & von Lilienfeld, 2020). As a result, TG80 exhibits *more diverse dynamics* and *improved numerical stability*, with no compound exceeding 50 Å center-of-mass drift in Figure 8[1].

---

[1]Simulations ran on 32 AMD EPYC 7543 cores with 256 GB RAM per molecule, totalling 806,400 CPU-hours (quoted market cost USD 150 000).

## 4 EXPERIMENT RESULTS

**Metrics.** We use *State-to-trajectory* (S2T) and *state-to-state* (S2S) error to evaluate ATOM (Xu et al., 2024). Specifically, S2T $= \frac{1}{P}\sum_{p=1}^{P}\|\hat{\mathbf{x}}_p - \mathbf{x}_p\|_2^2$, measures the average discrepancy between the predicted $\hat{\mathbf{x}}$ and ground-truth positions $\mathbf{x}$ across entire trajectories, while S2S $= \|\hat{\mathbf{x}}_P - \mathbf{x}_P\|_2^2$, quantifies the error at the final predicted timestep.

**Baselines.** For comparison, we include a range of classic to state-of-the-art baselines, including Radial Field (RF) (Köhler et al., 2019), Tensor Field Networks (TFN) (Thomas et al., 2018), SE(3) Transformer (SE(3)-Tr.) (Fuchs et al., 2020), E(n) equivariant graph neural networks (EGNN) (Garcia Satorras et al., 2021), MACE (Batatia et al., 2023), and EGNO (Xu et al., 2024). We note that MACE is pretrained on the authors' own dataset, so this is not a strictly like-for-like comparison. Our EGNN baselines are EGNN-Rollout (EGNN-R), which predicts timesteps autoregressively, and EGNN-Sequential (EGNN-S), which uses the output of each GNN as the prediction of a given frame. We set all baseline hyperparameters following previous works (Xu et al., 2024; 2022; Shi et al., 2021) and tune ATOM and EGNO hyperparameters as in Table 19 and Table 20.

**Training setups.** For training of ATOM and EGNO, we consider two temporal discretization strategies in selecting the timestamps $t_p = t + \sum_{r=1}^{p}\Delta t_r$: (1) *Uniform discretization* selects $t_p = t + p/P\Delta T$ and (2) *Tail discretization* selects $t_p = t + \overline{\Delta} + p/P(\Delta T - \overline{\Delta})$ for a lag $\overline{\Delta} \in [0, \Delta T]$. In the main paper, we present experiment results with uniform discretization and include the results with tail discretization in Appendix E. We perform early stopping on the lowest S2S validation loss checkpoint and report results as mean $\pm 2\sigma$ over *three training runs*. All experiments are run on an NVIDIA® RTX 5080 with wall-clock time and FLOP utilization detailed in Table 15.

### 4.1 SINGLE-TASK LEARNING

We benchmark on the MD17, RMD17, and MD22 DFT MD trajectory datasets (Chmiela et al., 2017; Christensen & von Lilienfeld, 2020; Chmiela et al., 2023). We partition the trajectories into train/validation/test splits of sizes 500/2000/2000, set $\Delta T = 3000$ fs and $P = 8$, and train for 2500 epochs following (Xu et al., 2024). For the performance on MD17 ( Table 1), we directly quote the results from (Xu et al., 2024) except for EGNO. We design ATOM to have six transformer blocks with a hidden size of 256.

**MD17 and RMD17.** As shown in Table 1, ATOM compares favorably with state-of-the-art (SOTA) baselines on MD17 dataset, yielding average reductions of 14.96% (S2S mean squared error (MSE)) and 8.3% (S2T MSE) on average[2]. In Table 9 (Appendix E.1), we benchmark ATOM on RMD17, and observe similarly competitive performance against EGNO.

**MD22.** To evaluate performance on larger molecules, we consider Ac-Ala3-NHMe (20 heavy atoms), docosahexaenoic acid (DHA with 24 heavy atoms), and stachyose (45 heavy atoms) from the MD22 dataset (Chmiela et al., 2023). ATOM remains competitive on these systems; whereas EGNO fails to converge (Table 2). We attribute this discrepancy to differing inductive biases: GNNs such as EGNO restrict message passing to a predefined bond or radius graph and can therefore under-represent long-range, non-bonded steric and electrostatic interactions that dominate the behavior of large, sparsely connected molecules (Alon &

Figure 2: Docosahexaenoic acid (DHA)

Yahav, 2021; Kosmala et al., 2023). This explains the poor performance of EGNO on MD22, which contains prototypically sparse molecules such as DHA, shown in Figure 2 (Nv et al., 2003). We further disentangle the role of connectivity from the use of attention by training a variant, ATOM-GATv2, in which our heterogeneous temporal attention is replaced by graph attention network v2 (GATv2) layers (Brody et al., 2022) operating on the same bond/radius graph as EGNO. ATOM-GATv2 still substantially underperforms the full ATOM model, indicating that the performance gains stem from the fully connected point-cloud interaction pattern rather than from attention alone.

---

[2]We exclude benzene from the table due to the previously discussed high numerical noise.

Table 1: Single-task MSE ($\times 10^{-2}$) on MD17. Upper part: S2S MSE. Lower part: S2T MSE.

| | Aspirin | Ethanol | Malonaldehyde | Naphthalene | Salicylic | Toluene | Uracil |
|---|---|---|---|---|---|---|---|
| RF | $10.94_{\pm 0.02}$ | $4.64_{\pm 0.02}$ | $13.93_{\pm 0.06}$ | $0.50_{\pm 0.02}$ | $1.23_{\pm 0.04}$ | $10.93_{\pm 0.08}$ | $0.64_{\pm 0.02}$ |
| TFN | $12.37_{\pm 0.36}$ | $4.81_{\pm 0.08}$ | $13.62_{\pm 0.16}$ | $0.49_{\pm 0.02}$ | $1.03_{\pm 0.04}$ | $10.89_{\pm 0.02}$ | $0.84_{\pm 0.04}$ |
| SE(3)-Tr. | $11.12_{\pm 0.12}$ | $4.74_{\pm 0.02}$ | $13.89_{\pm 0.04}$ | $0.52_{\pm 0.02}$ | $1.13_{\pm 0.04}$ | $10.88_{\pm 0.12}$ | $0.79_{\pm 0.04}$ |
| EGNN | $14.41_{\pm 0.30}$ | $4.64_{\pm 0.04}$ | $13.64_{\pm 0.02}$ | $0.47_{\pm 0.04}$ | $1.02_{\pm 0.04}$ | $11.78_{\pm 0.14}$ | $0.64_{\pm 0.02}$ |
| EGNN-R | $9.96_{\pm 0.14}$ | $4.61_{\pm 0.01}$ | $13.04_{\pm 0.03}$ | $0.44_{\pm 0.05}$ | $0.96_{\pm 0.01}$ | $10.19_{\pm 0.15}$ | $1.11_{\pm 0.04}$ |
| EGNN-S | $10.25_{\pm 0.09}$ | $4.61_{\pm 0.01}$ | $13.06_{\pm 0.04}$ | $0.53_{\pm 0.01}$ | $1.06_{\pm 0.05}$ | $10.83_{\pm 0.09}$ | $0.62_{\pm 0.01}$ |
| EGNO | $9.64_{\pm 0.15}$ | $4.57_{\pm 0.01}$ | $\mathbf{12.92}_{\pm 0.00}$ | $\mathbf{0.39}_{\pm 0.00}$ | $0.89_{\pm 0.01}$ | $11.00_{\pm 0.00}$ | $\mathbf{0.58}_{\pm 0.02}$ |
| MACE | $6.95_{\pm 0.00}$ | $2.06_{\pm 0.00}$ | $17.99_{\pm 0.26}$ | $0.72_{\pm 0.00}$ | $1.05_{\pm 0.00}$ | $6.44_{\pm 0.00}$ | $0.75_{\pm 0.00}$ |
| **ATOM** | $\mathbf{6.82}_{\pm 0.06}$ | $\mathbf{3.52}_{\pm 0.04}$ | $14.72_{\pm 0.01}$ | $0.50_{\pm 0.00}$ | $\mathbf{0.88}_{\pm 0.01}$ | $\mathbf{4.66}_{\pm 0.21}$ | $0.63_{\pm 0.00}$ |
| EGNN-R | $7.35_{\pm 0.19}$ | $3.21_{\pm 0.00}$ | $\mathbf{10.75}_{\pm 0.04}$ | $\mathbf{0.34}_{\pm 0.06}$ | $1.09_{\pm 0.12}$ | $4.53_{\pm 0.08}$ | $0.89_{\pm 0.02}$ |
| EGNN-S | $9.01_{\pm 0.34}$ | $3.21_{\pm 0.00}$ | $11.20_{\pm 0.03}$ | $0.42_{\pm 0.01}$ | $1.41_{\pm 0.00}$ | $4.86_{\pm 0.04}$ | $0.65_{\pm 0.01}$ |
| EGNO | $9.64_{\pm 0.15}$ | $4.57_{\pm 0.01}$ | $12.92_{\pm 0.00}$ | $0.39_{\pm 0.00}$ | $0.90_{\pm 0.01}$ | $10.99_{\pm 0.00}$ | $\mathbf{0.58}_{\pm 0.02}$ |
| MACE | $5.06_{\pm 0.00}$ | $2.84_{\pm 0.00}$ | $16.09_{\pm 0.03}$ | $0.57_{\pm 0.00}$ | $0.55_{\pm 0.00}$ | $3.26_{\pm 0.00}$ | $1.08_{\pm 0.00}$ |
| **ATOM** | $\mathbf{5.62}_{\pm 0.05}$ | $\mathbf{2.62}_{\pm 0.04}$ | $12.49_{\pm 0.01}$ | $0.43_{\pm 0.00}$ | $0.86_{\pm 0.01}$ | $2.27_{\pm 0.10}$ | $0.61_{\pm 0.00}$ |

Table 2: Single-task MSE ($\times 10^{-2}$) on MD22. Upper: S2S. Lower: S2T

| | Ac-Ala3-NHME | DHA | Stachyose |
|---|---|---|---|
| EGNO | $357.89_{\pm 3.94}$ | $178.39_{\pm 4.91}$ | $42.11_{\pm 0.10}$ |
| ATOM-GATv2 | $223.57_{\pm 0.66}$ | $16.72_{\pm 0.44}$ | $41.40_{\pm 0.37}$ |
| **ATOM** | $\mathbf{9.65}_{\pm 0.75}$ | $\mathbf{10.60}_{\pm 1.11}$ | $\mathbf{21.25}_{\pm 4.20}$ |
| Gap | +97.30% | +94.06% | +49.54% |
| EGNO | $232.40_{\pm 6.75}$ | $116.45_{\pm 3.34}$ | $30.84_{\pm 0.03}$ |
| ATOM-GATv2 | $113.26_{\pm 0.04}$ | $14.39_{\pm 0.32}$ | $29.70_{\pm 0.15}$ |
| **ATOM** | $\mathbf{7.55}_{\pm 0.42}$ | $\mathbf{9.66}_{\pm 1.16}$ | $\mathbf{18.13}_{\pm 3.78}$ |
| Gap | +96.75% | +91.70% | +41.22% |

Table 3: Multitask S2T MSE ($\times 10^{-2}$) on TG80 across five UMAP cluster assignments.

| | | Cluster 1 | Cluster 2 | Cluster 3 | Cluster 4 | Cluster 5 |
|---|---|---|---|---|---|---|
| ID | EGNO | $44.23_{\pm 0.68}$ | $95.52_{\pm 0.73}$ | $141.16_{\pm 0.21}$ | $150.92_{\pm 0.11}$ | $107.47_{\pm 0.36}$ |
| | **ATOM** | $\mathbf{9.71}_{\pm 0.75}$ | $\mathbf{18.26}_{\pm 1.58}$ | $\mathbf{16.82}_{\pm 1.46}$ | $\mathbf{16.93}_{\pm 3.65}$ | $\mathbf{17.20}_{\pm 0.46}$ |
| | Gap | 78.04% | 80.89% | 88.09% | 88.78% | 83.99% |
| OOD | MACE | 134.26 | 224.12 | 325.97 | 316.26 | 229.64 |
| | EGNO | $45.95_{\pm 0.80}$ | $115.43_{\pm 13.23}$ | $151.74_{\pm 0.57}$ | $163.90_{\pm 0.69}$ | $113.68_{\pm 2.50}$ |
| | EGNN-S | $45.44_{\pm 0.57}$ | $7386.15_{\pm 6931.89}$ | $152.72_{\pm 0.83}$ | $464.22_{\pm 509.48}$ | $114.30_{\pm 0.79}$ |
| | EGNN-R | $44.88_{\pm 0.68}$ | $109.62_{\pm 1.92}$ | $148.05_{\pm 0.70}$ | $161.54_{\pm 0.68}$ | $110.10_{\pm 0.96}$ |
| | **ATOM** | $\mathbf{35.05}_{\pm 0.97}$ | $\mathbf{106.99}_{\pm 104.64}$ | $\mathbf{60.95}_{\pm 4.86}$ | $\mathbf{66.68}_{\pm 0.96}$ | $\mathbf{47.49}_{\pm 1.59}$ |
| | Gap | 21.93% | 2.40% | 58.83% | 58.71% | 56.88% |

## 4.2 MULTITASK LEARNING ON TG80

We pretrain ATOM on TG80, scaling to six attention blocks with a hidden size of 256. We select stochastic horizons $\Delta T \sim \text{LogUnif}(8\,\text{fs}, 24\,000\,\text{fs})$ and use a five-fold, cluster-based cross-validation. Specifically, we compute ECFP-4 fingerprints (Rogers & Hahn, 2010), embed them using UMAP (McInnes et al., 2018), and apply agglomerative clustering (Ward, 1963) to partition compounds into ten disjoint clusters. The folds are then formed by holding out clusters, ensuring that the train/validation/test sets occupy distinct regions of chemical space. This cluster-wise protocol minimizes leakage and more closely reflects the prospective scientific setting in which models must generalize to unseen molecules. Cluster-based approaches present more challenging generalization problems than random splits or common chemical-scaffold-based splits (Guo et al., 2024). In Appendix E.2, we also consider pretraining on a standard random split of molecules.

Table 3 benchmarks ATOM by assessing both *in-distribution* (ID) and *out-of-domain* (OOD) S2T performance. For the *in-distribution* setting, we train, validate, and test on molecules from the same cluster. We observe that ATOM outperforms existing baselines by an average of 83.96% in terms of S2T MSE. We then assess *out-of-domain* (OOD) generalization performance by predicting the dynamics of unseen compounds drawn from disjoint clusters. Under OOD settings, ATOM nearly halves the S2T MSE of EGNO, with an average improvement of 39.74% across five cluster splits. Notably, OOD ATOM beats ID EGNO performance in four of five folds. This striking zero-shot generalization, realized without any exposure to the test molecules, confirms that ATOM uniquely learns robust, transferable knowledge of molecular dynamics. In Appendix E.2, we show similar outperformance in S2S prediction. In Appendix F.2, we show that the significantly improved multitask performance comes with a modest overhead in training time and in inference latency.

## 4.3 TEMPORAL GAP AND TIMESTEP INVARIANCE PROPERTIES

$\Delta T$ **Invariance.** We evaluate the performance of pretrained ATOM (at fixed $\Delta T = 3000$) with varying $\Delta T$ at inference. We compare ATOM, EGNO, and EGNN on S2T MSE by fixing $P = 8$ and sweeping $\Delta T$ logarithmically from 10 to 10 000 fs on an in-distribution (Cluster 1) multitask model. In Figure 3, we show that ATOM maintains its extrapolation advantage across the range compared to

EGNO, particularly at larger $\Delta T$. Ablating T-RoPE (NoPE) removes this advantage by exhibiting an EGNO-like error trend with substantially higher MSE. This underscores T-RoPE's role in stable time-gap extrapolation.

$P$ **Invariance.** Following the discretization invariance in neural operators, we expect ATOM and EGNO models to show consistent MSE as $P$ varies under uniform discretization (Kovachki et al., 2021). Figure 4 confirms such a conjecture by showing that multitask ATOM pretrained at $P = 8$ maintain constant S2T MSE as $P$ ranges from 4 to 24 at inference.

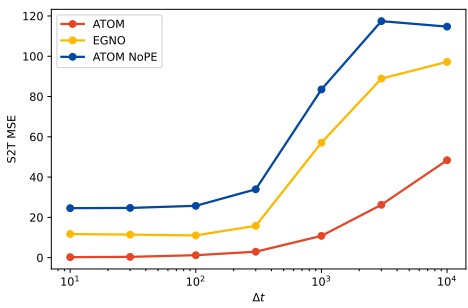
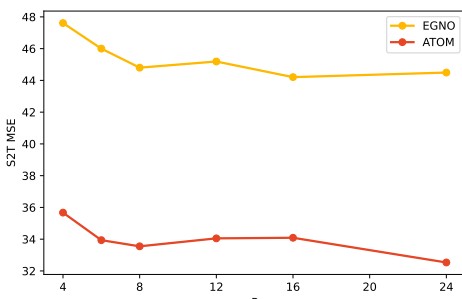

Figure 3: Pretrained ($\Delta T = 3000$, ID) multitask S2T MSE across varying $\Delta T$ values.

Figure 4: Pretrained ($P = 8$) ATOM and EGNO are discretization invariant, showing stable S2T MSE.

## 4.4 ABLATION STUDIES

We perform extensive ablations to assess each design choice in ATOM. For single-task performance (Fig. 5) and multitask performance (Fig. 6), we independently toggle components and measure their contributions. Our analysis focuses on equivariant lifting, T-RoPE, label-noise regularization, heterogeneous attention, and random-walk positional encoding (under multitask pretraining).

**Equivariant lifting.** We assess the quasi-equivariant design against a non-equivariant ATOM. As shown in Figure 5, replacing the equivariant lifting introduced in Section 3.2.1 with standard linear layers (no equivariant lifting) markedly degrades the performance of ATOM, increasing S2T MSE by 22.48. Table 4 further quantifies sensitivity to $SO(3)$ rotations: ATOM's S2T MSE increases by $10.80\times$ under rotation, compared to $19.77\times$ without equivariant lifting. Notably, the

Table 4: S2T MSE ($\times 10^{-2}$) of a fixed input frame rotated and unrotated by an $SO(3)$ matrix.

|  | ATOM | No equivariant Lift |
| --- | --- | --- |
| Unrotated | $6.76_{\pm 0.69}$ | $33.44_{\pm 23.42}$ |
| Rotated | $73.04_{\pm 27.01}$ | $660.97_{\pm 945.86}$ |
| Increase | $66.28_{\pm 26.32}$ | $627.53_{\pm 926.53}$ |
| Rotation penalty ($\times$) | 10.80 | 19.77 |

fully-equivariant variant of ATOM, described in appendix Appendix D.1, also underperforms ATOM in both single-task (Figure 5) and multitask (Figure 6) settings, with the gap exaggerated in the multitask setting. This aligns with recent findings on relaxed equivariance, suggesting that strict equivariance can limit model capacity and complicate the optimization process (Elhag et al., 2025). We present estimates of the quasi-equivariance $\varepsilon$ in Appendix E.3.

**Heterogeneous attention.** We find that substituting heterogeneous temporal attention with standard self-attention on the phase space features increases S2S MSE by 0.47, suggesting that cross-attention enables access to non-trivial feature interactions.

**Temporal Rotary Position Embedding (T-RoPE).** In the single-task regime with fixed $\Delta T$ (Figure 5), T-RoPE contributes little to the performance of ATOM, as it effectively reduces to a constant rotational shift. By contrast, with stochastic $\Delta T$, disabling T-RoPE (NoPE) increases MSE by 1.07, consistent with ATOM leveraging the $\tau$ parameter to encode variable time gaps (Figure 6). An EGNO-style sinusoidal positional encoding produces a similar performance degradation.

**Label noise regularization.** We also test the utility of label noise regularization as in Section 3.2.3. From Figure 5, we observe that removing augmented noise from the position and velocity features in-

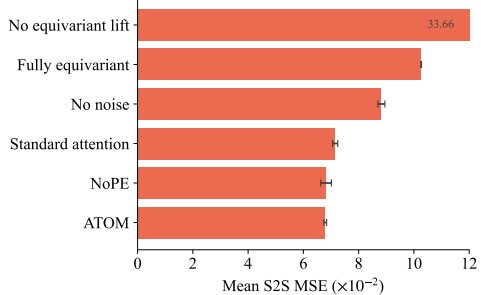

Figure 5: ATOM ablation on MD17 Aspirin.

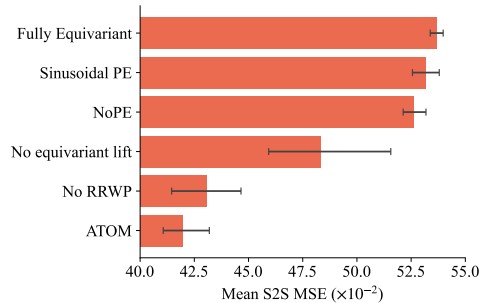

Figure 6: ATOM ablation on TG80 Cluster 1.

creased S2T MSE by 1.21. For the multitask ablation on TG80, we suppress label noise regularization, as the dataset is designed to be numerically stable with small noise.

**RWPE.** We assess random-walk positional encoding (RWPE) in the multitask pretraining. Figure 6 indicates that RWPE facilitates molecule identification, yielding improved multitask performance.

## 5 CONCLUSIONS

In this work, we demonstrate that carefully designed transformer neural operators enable zero-shot generalization to unseen chemical dynamics. Our experiments on MD17 demonstrate continued good single-task performance, and we present the first molecular neural operator that can successfully learn large molecule dynamics using MD22. Our multitask experiments show that our method learns transferable dynamics knowledge, even without explicit graph representations. In combination with our TG80 dataset, we provide a large-scale open-source benchmark and baselines to evaluate future models and spur further operator research with concrete scientific applicability.

**Limitations**   We remark that TG80 does not contain trajectories for large molecules with more than 15 heavy atoms, despite their obvious chemical and pharmacological relevance. In follow-up work, we intend to enrich TG80 with such molecules, calculated with a higher-resolution DFT basis set, $\omega$B97X-3c (Müller et al., 2023). Regarding ATOM, it lacks an explicit energy-based inductive bias, which may permit long-horizon drift. A natural extension is therefore a framewise energy head $E_\theta(\mathbf{x}_{t_p})$ with force supervision $\mathbf{F}_{t_p} = -\nabla_{\mathbf{x}_{t_p}} E_\theta(\mathbf{x}_{t_p})$. This energy term also defines the drift in the Langevin dynamics, where an additional stochastic term accounts for thermal perturbations of atomic positions. Incorporating such physics-informed stochastic dynamics into our operator learning framework is a natural next step, and we view this as a promising direction for future MD research.

## REPRODUCIBILITY STATEMENT

We provide experiment details, such as choice of hyperparameters and other training configurations in Appendix F. In addition, we will release the TG80 dataset upon acceptance under MIT license for reproducibility.

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

APPENDICES

## A    LLM ACKNOWLEDGMENT

The authors acknowledge the use of LLM for grammar corrections and for improving the clarity of the manuscript. The LLM was not used for generating original ideas, content, or experimental results. All conceptual contributions, analyses, and conclusions presented in this work are entirely from the authors.

## B    BACKGROUND

This section provides an introduction to the preliminaries of group theory.

### B.1    GROUPS

A group $(G, \circ)$ consists of a non-empty set $G$ and a binary operation $\circ : G \times G \to G$ satisfying the following axioms:

1. **Closure:** For all $a, b \in G$, the result of the operation $a \circ b$ is also in $G$: $a \circ b \in G$.
2. **Identity Element:** There exists an element $e \in G$ such that, for all $a \in G$, $a \circ e = e \circ a = a$.
3. **Associativity:** For all $a, b, c \in G$, $(a \circ b) \circ c = a \circ (b \circ c)$.
4. **Inverses:** For each $a \in G$, there exists an element $a^{-1} \in G$ such that $a \circ a^{-1} = a^{-1} \circ a = e$.

In general, not all groups are abelian. That is, the binary operation $\circ$ does not necessarily commute: $g \circ h = h \circ g, \forall g, h \in G$.

### B.2    GROUP REPRESENTATIONS

A group representation is a homomorphism $\rho : G \to GL(V)$ that assigns an $n \times n$ matrix to each group element $g \in G$, realizing it as a linear transformation. Representations must preserve the binary operation for all members of the group $G$ such that:

$$\rho(g \circ h) = \rho(g)\rho(h), \quad \forall g, h \in G.$$

A representation $\rho(g)$ is reducible if it can be represented as the direct sum of other representations:

$$\rho(g) = \rho_1(g) \oplus \rho_2(g), \quad \forall g \in G.$$

For example, a reducible $4 \times 4$ representation of $SU(2)$ can be decomposed into two $2 \times 2$ sub-representations:

$$\rho(g) = \begin{bmatrix} \rho_1(g) & 0 \\ 0 & \rho_2(g) \end{bmatrix}, \quad \forall g \in SU(2),$$

where $\rho_1(g)$ and $\rho_2(g)$ are the following irreducible representations of $SU(2)$:

$$\rho_1(g) = \begin{bmatrix} e^{i\theta} & 0 \\ 0 & e^{-i\theta} \end{bmatrix}, \quad \rho_2(g) = \begin{bmatrix} e^{i\phi} & 0 \\ 0 & e^{-i\phi} \end{bmatrix}.$$

By contrast, irreducible representations or *irreps* cannot be represented as such a direct sum. Formally, they have no non-trivial invariant subspaces $W \subset V$ such that $\rho(g)W \subset W, \forall g \in G$.

Representing inputs as irreps ensures equivariance by constraining each feature to transform predictably under group actions. Given $V = \bigoplus_i V_i$ with irreps $V_i$, the transformation of an input $x \in V$ under $g \in G$ is:

$$\rho(g)x = \bigoplus_i \rho_i(g)x_i.$$

Each component $x_i$ transforms independently according to $\rho_i$, preserving symmetry. Scalars remain invariant, while vectors rotate according to standard representations. This decomposition prevents the mixing of differently transforming features, ensuring that all subsequent operations, linear or non-linear, respect the group's symmetry, thereby maintaining equivariance throughout the network.

Intuitively, the tensor products capture interactions between features in a manner akin to multiplication, producing a higher-dimensional representation. Crucially, this new representation is reducible, so we may decompose it into irreps:

$$V \otimes V \cong \bigoplus_k V_k.$$

It is this decomposition that allows the network to project onto individual irreps, achieving non-trivial feature mixing whilst preserving symmetry constraints.

## C  DATASETS

We present a visualization of a sample trajectory of uracil from three datasets in Figure 7.

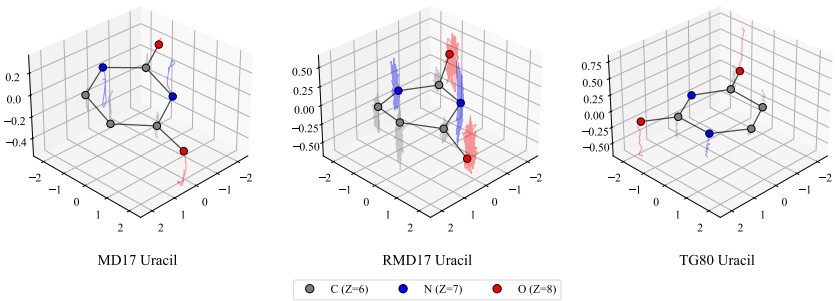

Figure 7: 3000 timesteps of uracil trajectory from MD17, RMD17, and TG80.

### C.1  LICENCES

Table 5: Dataset sources and licenses. We release TG80 under the MIT license.

| Dataset | Source | License |
|---------|--------|---------|
| MD17  | https://www.sgdml.org/ | CC BY 4.0 |
| RMD17 | https://archive.materialscloud.org/record/2020.82 | CC Zero V1.0 Universal |
| MD22  | https://www.sgdml.org/ | CC BY 4.0 |
| TG80  | To be released at URL | MIT |

### C.2  MODEL INPUTS AND THE DATALOADER

Our compound representations follow (Shi et al., 2021; Xu et al., 2024). We model hydrogen atoms implicitly and concatenate the position and velocity norms for each node $i$ with their respective vectors. Unlike their implementations, we avoid explicit graph construction and do not include edge labels describing atomic bond geometries.

We duplicate all frames $\mathcal{G}^{(t)} \to \{\mathcal{G}^{(t)}\}^P$ during dataset initialization, producing a five-fold improvement in throughput compared to previous dataloaders in Table 6.

Table 6: Mean time (seconds) to produce $10\,000$ batches over 100 benchmark runs. Batch size = 100, 500 samples, $\Delta t = 3000$, 500 warmup batches.

| | Aspirin | Ethanol | Naphthalene | Toluene |
|---|---------|---------|-------------|---------|
| EGNO | $0.060_{\pm 0.024}$ | $0.024_{\pm 0.016}$ | $0.056_{\pm 0.024}$ | $0.039_{\pm 0.024}$ |
| ATOM | $\mathbf{0.005}_{\pm 0.002}$ | $\mathbf{0.007}_{\pm 0.002}$ | $\mathbf{0.008}_{\pm 0.004}$ | $\mathbf{0.006}_{\pm 0.002}$ |

## C.3 NUMERICAL STABILITY

We evaluate the numerical stability of MD17, RMD17, and TG80. MD17 benzene exhibits substantial center-of-mass drift in Figure 8a, which is also partially visible in the consistent motion trails shown in Figure 10a. RMD17 exhibits improved stability, with no center-of-mass drift exceeding $1 \times 10^4$. TG80 shows the lowest drift of all datasets, and expectedly includes more molecules with high per-step drift (due to more complex sterically hindered geometries).

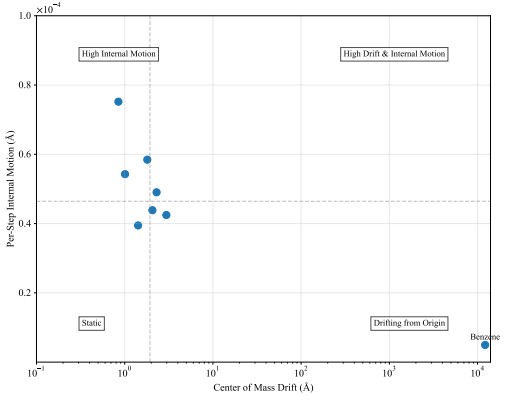

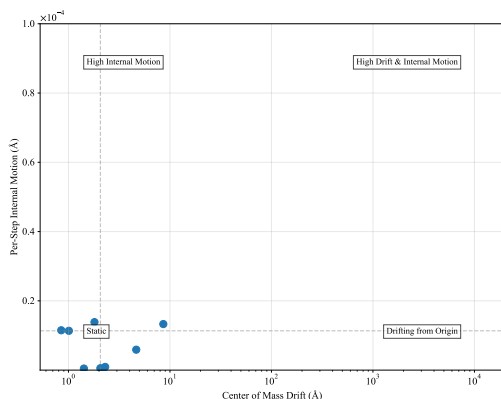

(a) MD17 molecules are largely consistent, except for benzene, which exhibits substantial drift.

(b) RMD17 molecules are more numerically stable, supporting their use in future benchmarks.

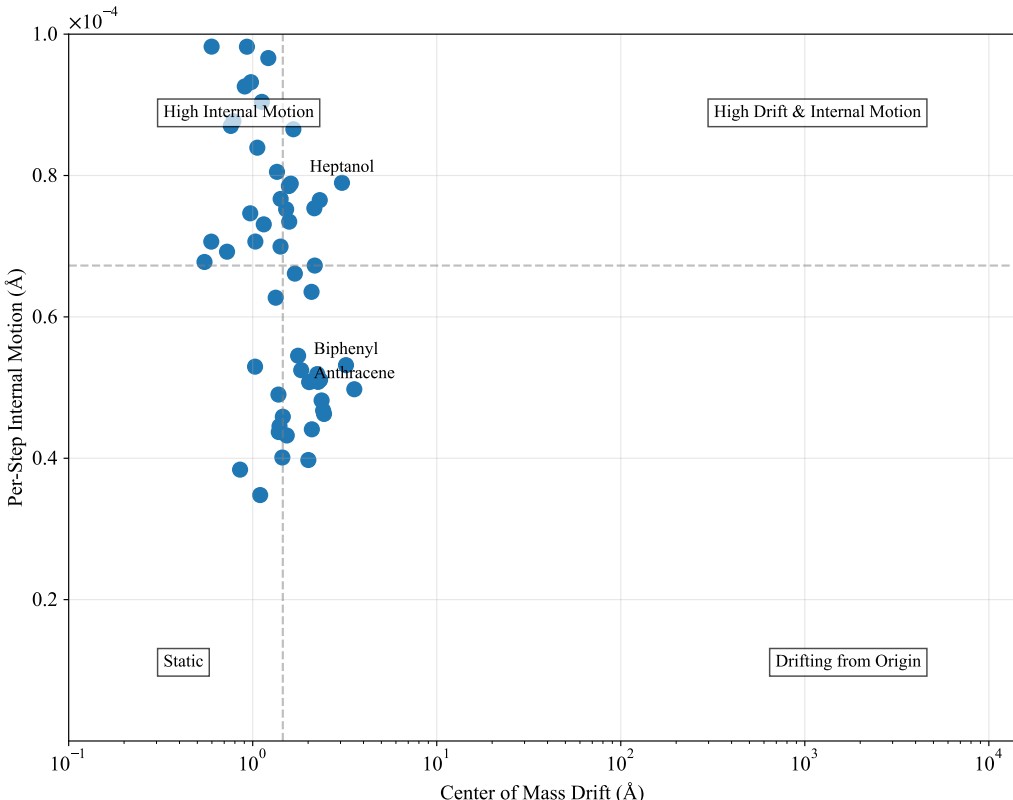

(c) TG80 molecules exhibit the lowest centre-of-mass drift among the evaluated MD datasets.

Figure 8: Comparison of numerical stability across MD17, RMD17, and TG80 datasets. Dashed lines denote the mean centre-of-mass drift and per-step motion; datapoints exceeding two standard deviations are annotated.

### C.4  TG80 GENERATION ALGORITHM

We first recall the definition of Tanimoto T similarity between two bit vectors $X, Y$ as

$$T(X, Y) = \frac{|X \cap Y|}{|X \cup Y|},$$

which is identical to the definition of the Jaccard similarity in this case (Rogers & Tanimoto, 1960).

To generate TG80, we randomly shuffled the PubChem dataset, then iterated through all compounds until 40 were found that matched the following criteria:

1. Simplified Molecular-input Line-entry System (SMILES) encode a valid molecular structure

2. No more heavy atoms than the corresponding seed molecule

3. Only contain {C, H, O, N} atoms

4. No more than five oxygen atoms

5. No more than three nitrogen atoms

6. No disconnected molecular fragments (e.g., salts)

7. Tanimoto similarity to at least one seed molecule greater than 0.875, less than 0.925

8. Tanimoto similarity to a previously selected molecule is no more than 0.2

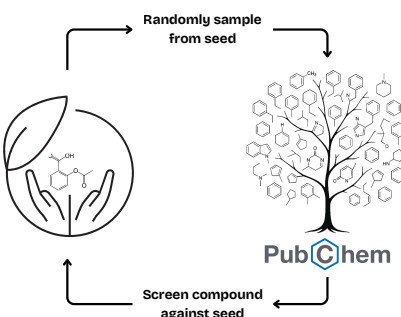

Figure 9: Construction of TG80 from an initial seed using the PubChem database.

This controlled selection procedure generates structurally analogous subsets around each seed molecule whilst preventing convergence to highly similar molecules across different seed groups.

Only 2,488 of the 173 million in the PubChem library satisfied the filtration criteria above. This low yield largely reflects the cumulative effect of criterion 8: as more molecules are added, it becomes harder to find candidates sufficiently dissimilar to all prior selections. Given that the average Tanimoto similarity to our seed set was just 0.1492, the 0.875 threshold was highly selective. Dataset generation code is available at ANONYMIZED.

### C.5  MOLECULAR DYNAMICS SIMULATIONS

We present a complete overview of the DFT parameters used to generate MD17 (Chmiela et al., 2017), RMD17 (Christensen & von Lilienfeld, 2020), MD22 (Chmiela et al., 2023), and TG80.

Table 7: An overview of the methodologies used to generate the MD datasets featured.

|  | DFT Functional | Dispersion Corrections | Basis set | Timestep | Temperature |
|---|---|---|---|---|---|
| MD17 | PBE | TS | NAO | 0.5 fs | 500K |
| RMD17 | PBE | None | def2-SVP | 0.5 fs | 500K |
| MD22 | PBE | MBD | NAO | 1.0 fs | 500K |
| TG80 | PBE | $\Delta 4$ | def2-SVP | 1.0 fs | 300K |

# D   ARCHITECTURAL DETAILS

In this section, we discuss the finer architectural details of ATOM and the architectural adjustments made for our ablations.

## D.1   FULLY EQUIVARIANT ATOM

To achieve the full equivariance discussed in Figure 5, we employ a canonicalization network approach, which removes Euclidean gauge before learning and then reinstates it afterwards (Kaba et al., 2023). This preserves equivariance of the whole network, even with the use of non-equivariant architectures in the trunk. This provides a controlled comparison between equivariant and non-equivariant ATOM: it enforces equivariance by construction while otherwise keeping the core architecture unchanged.

We first make data translation equivalent by centering

$$\mu = \frac{1}{N} \sum_{i=1}^{N} x_i, \qquad \bar{x}_i = x_i - \mu. \tag{2}$$

We then remove rotations by aligning to the second moment

$$S = \frac{1}{N} \sum_{i=1}^{N} \bar{x}_i \bar{x}_i^\top = \sum_{k=1}^{3} \lambda_k e_k e_k^\top \quad (\lambda_1 \geq \lambda_2 \geq \lambda_3), \tag{3}$$

and choose $e_1$ as the principal axis and orthonormalise

$$e_2 \leftarrow \frac{e_2 - (e_2^\top e_1)e_1}{\|e_2 - (e_2^\top e_1)e_1\|}, \qquad e_3 = e_1 \times e_2. \tag{4}$$

We can then form $Q = [e_1, e_2, e_3] \in \mathrm{SO}(3)$ and canonicalise

$$\tilde{x}_i = (x_i - \mu)Q, \qquad \tilde{v}_i = v_i Q. \tag{5}$$

We fix the eigenvector sign ambiguity using the chirality pseudoscalar $c_0 = \sum_{i=1}^{N} x_i \times v_i$ at the reference time (flip $e_1$ to satisfy $e_1^\top c_0 \geq 0$, then adjust $e_2, e_3$ jointly to keep right-handedness). Let $F$ be an arbitrary trunk acting in the canonical frame; with per-atom canonical outputs $\hat{y}_i = F(\{\tilde{x}_j, \tilde{v}_j\}_{j=1}^{N})_i$, we decanonicalise by

$$y_i = \hat{y}_i Q^\top + \mu. \tag{6}$$

This results in exact SE(3)-equivariance (Kaba et al., 2023) and permits non-equivariant trunks.

## D.2   RANDOM-WALK POSITIONAL ENCODINGS

In the multitask case, we add row-normalized random walk positional encoding (RWPE) to equip ATOM and EGNO with multiscale connectivity features, enhancing their ability to distinguish non-isomorphic graphs (Dwivedi et al., 2022; Ma et al., 2023). We first form a $\varepsilon$-neighborhood graph from our pointclouds as:

$$G = (V, E), \quad V = \{i\}, \quad E = \{(i,j) : \|(x,y,z)_i - (x,y,z)_j\|_2 < \varepsilon\}. \tag{7}$$

We set $\varepsilon = 1.6$, as covalent bonds typically range from 1.14 Å to 2.0 Å in length (Lobato et al., 2021) and highlight that this construction does not necessitate prior knowledge of the graph structure.

Let $\mathbf{A} \in \mathbb{R}^{n \times n}$ denote the adjacency matrix of this graph, and let $\mathbf{D} = \mathrm{diag}(\mathbf{A1})$ represent its degree matrix. We construct the random walk transition matrix as $\mathbf{M} = \mathbf{D}^{-1}\mathbf{A}$ then compute matrix powers of $\mathbf{M}$ up to a maximum walk-length $K$, defining the self-return probabilities for each node as

$$p_i^{(k)} = \left(\mathbf{M}^k\right)_{ii}, \quad k = 1, \dots, K. \tag{8}$$

These probabilities are collected into vectors $\mathbf{p}_i \in \mathbb{R}^K$ and concatenated with the phase space to form $\hat{\mathbf{z}} = (\mathbf{v} \parallel \mathcal{Z} \parallel \mathbf{p}) \in W_{\mathrm{in}}$. Here, the input feature space is redefined as $W_{\mathrm{in}} = V_{\mathrm{in}} \oplus \rho_0^{\mathrm{even}} \oplus \left(\rho_0^{\mathrm{even}} \otimes \mathbb{R}^K\right)$, and the subsequent equivariant maps are modified in kind.

## D.3 Value-residual Learning

We employ value-residual learning wherein each transformer block receives the output of the first block via a residual connection to stabilize training and information flow through the network (Zhou et al., 2024b). Inspired by (Jordan, 2024), we add a learned coefficient to weight this residual. Here, $v$ denotes the current block's value output, and $v_1$ represents the initial block's value. A learnable parameter $\alpha$ is passed through a sigmoid to obtain the weighting coefficient:

$$\lambda = \sigma(\alpha). \tag{9}$$

The combined output is then given by:

$$v = \lambda\, v + (1 - \lambda)\, v_1. \tag{10}$$

In practice, we lock the first block's $\lambda$ value to $0.5$.

## D.4 Delta-prediction

When delta-prediction is enabled, as in Figure 5, we incorporate the initial positions $\mathbf{x}$ as a residual term, reformulating the model as an operator that learns a displacement field rather than predicting absolute positions. We express this as:

$$\mathbf{x}^{\dagger} = \mathrm{Project}(\mathbf{x}_{\mathrm{out}}) + \mathbf{x}. \tag{11}$$

Although this approach is implemented in both EGNN and EGNO, we found it was disabled by default in the codebase of the latter (Garcia Satorras et al., 2021; Xu et al., 2024). Based on empirical results from our ablations, Figure 5, we argue there is sufficient evidence to discourage the use of delta-prediction in neural operator-based molecular dynamics simulations.

## E Further Experiments

We conduct further experiments on single-task and multitask learning. We consider performance under tail discretization and report results on the RMD17 dataset. For multitask learning, we report performance under random cluster assignment and S2S metrics for the experiments in Section 4.2.

### E.1 Further Single-task Learning Experiments

**MD17 with tail discretization** We find the performance of both EGNO and ATOM on MD17 with tail discretization remains similar to the performance under uniform discretization discussed in table 1.

Table 8: EGNO and ATOM with final frame sampling. Upper part: S2S MSE. Lower part: S2T MSE.

|  | Aspirin | Benzene | Ethanol | Malonaldehyde | Naphthalene | Salicylic | Toluene | Uracil |
|---|---|---|---|---|---|---|---|---|
| EGNO | $9.66_{\pm0.12}$ | $39.09_{\pm2.35}$ | $4.57_{\pm0.01}$ | $\mathbf{12.92}_{\pm0.00}$ | $0.39_{\pm0.00}$ | $0.88_{\pm0.01}$ | $10.99_{\pm0.00}$ | $0.60_{\pm0.00}$ |
| ATOM | $\mathbf{6.38}_{\pm0.17}$ | $\mathbf{39.03}_{\pm3.32}$ | $\mathbf{3.62}_{\pm0.08}$ | $15.26_{\pm0.65}$ | $0.39_{\pm0.00}$ | $\mathbf{0.83}_{\pm0.01}$ | $\mathbf{5.26}_{\pm0.79}$ | $\mathbf{0.55}_{\pm0.00}$ |
| Gap | $+33.97\%$ | $+0.15\%$ | $+20.85\%$ | $-18.06\%$ | $+1.62\%$ | $+4.75\%$ | $+52.13\%$ | $+9.28\%$ |
| EGNO | $9.66_{\pm0.11}$ | $39.15_{\pm2.28}$ | $4.57_{\pm0.01}$ | $\mathbf{12.92}_{\pm0.01}$ | $0.39_{\pm0.00}$ | $0.88_{\pm0.01}$ | $10.99_{\pm0.00}$ | $0.60_{\pm0.00}$ |
| ATOM | $\mathbf{6.38}_{\pm0.17}$ | $\mathbf{39.03}_{\pm3.35}$ | $3.63_{\pm0.08}$ | $15.21_{\pm0.60}$ | $\mathbf{0.38}_{\pm0.00}$ | $\mathbf{0.83}_{\pm0.01}$ | $\mathbf{5.27}_{\pm0.79}$ | $\mathbf{0.55}_{\pm0.00}$ |
| Gap | $+33.91\%$ | $+0.30\%$ | $+20.66\%$ | $-17.71\%$ | $+1.82\%$ | $+5.02\%$ | $+52.08\%$ | $+9.44\%$ |

**Revised MD17 Dataset** We reach performance parity with EGNO on RMD17, shown in Table 9.

Table 9: EGNO and ATOM with final frame sampling. Upper part: S2S MSE. Lower part: S2T MSE.

|  | Azobenzene | Ethanol | Malonaldehyde | Naphthalene | Paracetamol | Salicylic | Toluene | Uracil |
|---|---|---|---|---|---|---|---|---|
| EGNO | $8.96_{\pm0.03}$ | $\mathbf{23.26}_{\pm0.01}$ | $\mathbf{40.11}_{\pm0.05}$ | $1.42_{\pm0.00}$ | $\mathbf{28.08}_{\pm0.01}$ | $1.06_{\pm0.01}$ | $28.28_{\pm0.01}$ | $0.88_{\pm0.00}$ |
| ATOM | $\mathbf{8.88}_{\pm0.05}$ | $23.49_{\pm0.14}$ | $40.29_{\pm0.13}$ | $\mathbf{1.36}_{\pm0.00}$ | $30.12_{\pm0.87}$ | $\mathbf{1.03}_{\pm0.00}$ | $28.56_{\pm0.04}$ | $\mathbf{0.86}_{\pm0.00}$ |
| Gap | $+0.90\%$ | $-0.99\%$ | $-0.45\%$ | $+3.93\%$ | $-7.26\%$ | $+3.10\%$ | $-0.99\%$ | $+1.90\%$ |
| EGNO | $8.51_{\pm0.03}$ | $\mathbf{23.61}_{\pm0.03}$ | $\mathbf{40.32}_{\pm0.08}$ | $1.42_{\pm0.00}$ | $\mathbf{28.01}_{\pm0.02}$ | $1.07_{\pm0.01}$ | $28.23_{\pm0.00}$ | $0.87_{\pm0.00}$ |
| ATOM | $\mathbf{8.38}_{\pm0.05}$ | $23.90_{\pm0.15}$ | $40.67_{\pm0.17}$ | $\mathbf{1.36}_{\pm0.00}$ | $30.03_{\pm0.78}$ | $\mathbf{1.04}_{\pm0.00}$ | $28.58_{\pm0.05}$ | $\mathbf{0.85}_{\pm0.00}$ |
| Gap | $+1.47\%$ | $-1.27\%$ | $-0.88\%$ | $+4.39\%$ | $-7.21\%$ | $+2.78\%$ | $-1.23\%$ | $+2.00\%$ |

### E.2 FURTHER MULTITASK LEARNING EXPERIMENTS

**Random-split cross-validation on TG80.** For completeness, we report multitask results under compound-level random cross-validation, where compounds are randomly assigned to the train, validation, and test sets. Relative to the more challenging out-of-domain (UMAP-based) split in Table 3, EGNO is comparatively stronger; nevertheless, ATOM maintains a consistent lead across folds, with mean improvements of 24.43% on S2S and 23.93% on S2T.

Table 10: S2S MSE ($\times 10^{-2}$) on TG80 across five random cluster assignments.

|  |  | Cluster 1 | Cluster 2 | Cluster 3 | Cluster 4 | Cluster 5 |
|---|---|---|---|---|---|---|
| OOD | EGNO | $71.83_{\pm 0.00}$ | $76.92_{\pm 0.00}$ | $68.99_{\pm 0.00}$ | $101.27_{\pm 0.00}$ | $83.20_{\pm 0.00}$ |
|  | ATOM | $\mathbf{53.93}_{\pm 0.00}$ | $\mathbf{62.40}_{\pm 0.00}$ | $\mathbf{49.37}_{\pm 0.00}$ | $\mathbf{70.75}_{\pm 0.00}$ | $\mathbf{66.75}_{\pm 0.00}$ |
|  | Gap | $+24.92\%$ | $+18.88\%$ | $+28.45\%$ | $+30.14\%$ | $+19.77\%$ |

Table 11: S2T MSE ($\times 10^{-2}$) on TG80 across five random cluster assignments.

|  |  | Cluster 1 | Cluster 2 | Cluster 3 | Cluster 4 | Cluster 5 |
|---|---|---|---|---|---|---|
| OOD | EGNO | $63.23_{\pm 0.00}$ | $64.49_{\pm 0.00}$ | $59.18_{\pm 0.00}$ | $85.87_{\pm 0.00}$ | $69.46_{\pm 0.00}$ |
|  | ATOM | $\mathbf{46.09}_{\pm 0.00}$ | $\mathbf{54.47}_{\pm 0.00}$ | $\mathbf{42.90}_{\pm 0.00}$ | $\mathbf{55.64}_{\pm 0.00}$ | $\mathbf{59.55}_{\pm 0.00}$ |
|  | Gap | $+27.10\%$ | $+15.54\%$ | $+27.51\%$ | $+35.21\%$ | $+14.28\%$ |

**Multitask S2S results on TG80 under UMAP cluster cross-validation.** The S2S side of the multitask learning results follow closely from their S2T counterparts presented in Section 4.2.

Table 12: S2S MSE ($\times 10^{-2}$) on TG80 across five UMAP cluster assignments.

|  |  | Cluster 1 | Cluster 2 | Cluster 3 | Cluster 4 | Cluster 5 |
|---|---|---|---|---|---|---|
| ID | EGNO | $51.98_{\pm 0.81}$ | $95.86_{\pm 0.53}$ | $142.51_{\pm 0.58}$ | $155.25_{\pm 0.67}$ | $109.25_{\pm 0.24}$ |
|  | ATOM | $\mathbf{15.49}_{\pm 1.04}$ | $\mathbf{26.55}_{\pm 2.13}$ | $\mathbf{28.74}_{\pm 2.40}$ | $\mathbf{29.81}_{\pm 2.72}$ | $\mathbf{26.33}_{\pm 1.98}$ |
|  | Gap (%) | $70.20\%$ | $72.30\%$ | $79.83\%$ | $80.80\%$ | $75.90\%$ |
| OOD | EGNO | $52.90_{\pm 0.72}$ | $114.14_{\pm 13.21}$ | $149.99_{\pm 0.34}$ | $163.47_{\pm 1.00}$ | $112.36_{\pm 1.90}$ |
|  | EGNN-S | $52.39_{\pm 0.40}$ | $16512.07_{\pm 12314.09}$ | $149.41_{\pm 0.94}$ | $663.54_{\pm 865.23}$ | $111.08_{\pm 0.62}$ |
|  | EGNN-R | $52.08_{\pm 0.79}$ | $\mathbf{108.89}_{\pm 1.60}$ | $148.67_{\pm 0.73}$ | $163.27_{\pm 0.16}$ | $109.94_{\pm 0.31}$ |
|  | ATOM | $\mathbf{41.97}_{\pm 1.24}$ | $127.95_{\pm 122.67}$ | $\mathbf{74.53}_{\pm 4.82}$ | $\mathbf{80.95}_{\pm 1.21}$ | $\mathbf{58.26}_{\pm 1.68}$ |
|  | Gap (%) | $19.41\%$ | $-17.50\%$ | $49.87\%$ | $50.42\%$ | $47.01\%$ |

**Multitask S2S Versus Single Task** We evaluate a multitask ATOM model trained on all available trajectories, pooling both ID and OOD clusters, against a single-task ATOM trained separately on each compound. The multitasking model achieves losses comparable to or lower than those of the single-task baselines, despite being trained with the same compute resources.

### E.3 Monte Carlo Estimation of Quasi-equivariance

For both the pretrained ATOM and the non-equivariant lifting variant, we estimate the quantity in Section 3.1 by Monte Carlo, drawing $N$ random timesteps $x_n$ and, for each, $R$ random rotations $g_{n,r} \in G$. We approximate

Table 13: MSE $(\times 10^{-2})$ on full-dataset TG80 ATOM and single-task ATOM. S2S upper, S2T lower.

|  | Formic Acid |
| --- | --- |
| Single-task ATOM | 26.40 |
| All-data ATOM | **22.39** |
| Gap (%) | 15.19% |
| Single-task ATOM | **18.10** |
| All-data ATOM | 18.72 |
| Gap (%) | −3.43% |

$$\varepsilon \approx \frac{1}{N} \sum_{n=1}^{N} \left\| \frac{1}{R} \sum_{r=1}^{R} \big( f\big(\phi(g_{n,r})(x_n)\big) - \rho(g_{n,r})\big(f(x_n)\big) \big) \right\|_2.$$

We report our estimates of $\varepsilon$ for various ATOM models trained on MD17 single task learning in Table 18. In all cases, ATOM shows a substantially lower equivariance defect, supporting our claim that our quasi-equivariant design achieves a kind of middle-ground in the trade-off between expressiveness and strict equivariance.

Table 14: Estimates of Quasi-equivariance $\varepsilon$ via Monte Carlo over 20 rotations and 10 timesteps with 2SD intervals.

|  | Aspirin | Ethanol | Malonaldehyde | Naphthalene | Salicylic | Toluene | Uracil |
| --- | --- | --- | --- | --- | --- | --- | --- |
| Non-equivariant ATOM | $120.23_{\pm153.71}$ | $102.50_{\pm45.14}$ | $26.94_{\pm16.04}$ | $37.97_{\pm12.65}$ | $37.81_{\pm20.08}$ | $57822.95_{\pm42872.97}$ | $32.29_{\pm35.80}$ |
| **ATOM** | $\mathbf{28.34}_{\pm25.10}$ | $\mathbf{11.09}_{\pm9.64}$ | $\mathbf{9.99}_{\pm9.82}$ | $\mathbf{17.09}_{\pm13.03}$ | $\mathbf{25.46}_{\pm21.47}$ | $\mathbf{2744.07}_{\pm3108.52}$ | $\mathbf{28.72}_{\pm19.20}$ |

## F Experimental Details

### F.1 Software and Hardware Details

All experiments were conducted using Python 3.12, NumPy 2.2.1 (Harris et al., 2020), PyTorch 2.5.1 (Paszke et al., 2019), e3nn 0.5.6 (Geiger & Smidt, 2022) and PyTorch Optimizer 3.5.0 (Kim, 2021). We use RDKit 2024.9.6 (Landrum et al., 2025) and PubChemPy 1.0.4 to construct TG80. All single-task training was performed on an NVIDIA RTX 5080 (16 GB) with CUDA 12.4, running on Ubuntu 24.04. We use Ase 3.26.0 (Larsen et al., 2017) and MACE-Torch 0.3.14 (Kovács et al., 2025) in the experiments of Appendix F.3.

### F.2 Training Times and Compute Requirements

**Single-task training time** We roughly wall-clock normalised our ATOM and EGNO parameter counts, resulting in respective learnable parameter counts of 754 468 and 335 770.

Table 15: Compute cost of single-task training on all MD17 molecules over 1000 epochs. Both ATOM (335 770 params) and EGNO (754 468 params) are under `torch.compile` on a Titan V.

| Model | Metric | Azobenzene | Ethanol | Malonaldehyde | Naphthalene | Paracetamol | Salicylic | Toluene | Uracil |
| --- | --- | --- | --- | --- | --- | --- | --- | --- | --- |
| EGNO | Time (mins) | $4.09_{\pm0.15}$ | $3.62_{\pm0.42}$ | $3.65_{\pm0.01}$ | $5.01_{\pm0.02}$ | $9.02_{\pm1.22}$ | $5.16_{\pm0.03}$ | $3.93_{\pm0.02}$ | $4.35_{\pm0.04}$ |
|  | Total FLOPS $(\times10^{12})$ | 3681.24 | 3257.70 | 3282.09 | 4513.37 | 8114.44 | 4641.50 | 3539.92 | 3915.98 |
|  | Epochs/min | 244.48 | 276.27 | 274.22 | 199.41 | 110.91 | 193.90 | 254.24 | 229.83 |
| ATOM | Time (mins) | $5.81_{\pm0.02}$ | $5.79_{\pm0.06}$ | $5.79_{\pm0.00}$ | $5.86_{\pm0.01}$ | $5.89_{\pm0.02}$ | $5.85_{\pm0.01}$ | $5.81_{\pm0.01}$ | $5.83_{\pm0.02}$ |
|  | Total FLOPS $(\times10^{12})$ | 5226.49 | 5212.53 | 5213.50 | 5271.19 | 5297.84 | 5263.76 | 5224.91 | 5247.33 |
|  | Epochs/min | 172.20 | 172.66 | 172.63 | 170.74 | 169.88 | 170.98 | 172.25 | 171.52 |
| Total FLOPS Reduction (%) |  | −41.98% | −60.01% | −58.85% | −16.79% | +34.71% | −13.41% | −47.60% | −34.00% |

**Multitask training time** In multitask training on TG80, our upsized ATOM model contained 3 557 840 parameters, compared to 335 770 for EGNO. Despite this, ATOM only trained between 5% and 30% slower than EGNO. This is perhaps unsurprising given the much higher FLOPS-utilization of the transformer architecture upon which ATOM is based.

Table 16: Compute cost of single task training on five TG80 molecules over 1000 epochs. Both ATOM (335 770 params) and EGNO (754 468 params) are under `torch.compile` on a Titan V.

| Model | | Fold1 | Fold2 | Fold3 | Fold4 | Fold5 |
|---|---|---|---|---|---|---|
| EGNO | Time (mins) | $9.61_{\pm1.21}$ | $8.56_{\pm0.06}$ | $9.04_{\pm0.11}$ | $9.31_{\pm0.20}$ | $8.98_{\pm0.01}$ |
| | Total FLOPS ($\times10^{12}$) | 8645.66 | 7703.33 | 8136.98 | 8378.06 | 8084.98 |
| | Epochs/min | 104.10 | 116.83 | 110.61 | 107.42 | 111.32 |
| ATOM | Time (mins) | $10.16_{\pm0.49}$ | $10.55_{\pm0.02}$ | $11.62_{\pm0.41}$ | $12.38_{\pm0.12}$ | $10.39_{\pm0.41}$ |
| | Total FLOPS ($\times10^{12}$) | 9140.57 | 9497.79 | 10455.34 | 11141.73 | 9355.37 |
| | Epochs/min | 98.46 | 94.76 | 86.08 | 80.78 | 96.20 |
| Total FLOPS Reduction (%) | | $-5.72\%$ | $-23.29\%$ | $-28.49\%$ | $-32.99\%$ | $-15.71\%$ |

### F.3 INFERENCE TIMES

We compare inference times on MD17 and MD22 across ATOM, the pretrained machine learning interaction potential MACE-OFF24 (Medium) (Kovács et al., 2025), and the classical Lennard-Jones potential (Larsen et al., 2017; Schwerdtfeger & Wales, 2024) and the molecular forcefield. We report inference times in seconds with 2SD intervals. We exclude AMBER results on Uracil as we were unable to run simulations for this molecule (Case et al., 2023).

Table 17: Seconds to produce timestep $\Delta T = 3000$ of each MD17 trajectory at `float32` precision.

| | | Aspirin | Ethanol | Malonaldehyde | Naphthalene | Salicylic | Toluene | Uracil |
|---|---|---|---|---|---|---|---|---|
| $\Delta T = 3000$ | MACE-OFF | $42.605_{\pm3.945}$ | $39.656_{\pm0.568}$ | $39.782_{\pm0.141}$ | $39.916_{\pm0.155}$ | $39.620_{\pm0.030}$ | $40.271_{\pm2.851}$ | $40.166_{\pm0.902}$ |
| | AMBER | $37.746_{\pm0.536}$ | $38.512_{\pm1.243}$ | $36.806_{\pm0.194}$ | $38.512_{\pm1.243}$ | $37.621_{\pm1.412}$ | $38.061_{\pm0.848}$ | – |
| | Lennard-Jones | $2.499_{\pm0.266}$ | $1.604_{\pm0.208}$ | $1.529_{\pm0.039}$ | $2.174_{\pm0.094}$ | $1.981_{\pm0.011}$ | $1.906_{\pm0.006}$ | $1.804_{\pm0.196}$ |
| | ATOM | $0.849_{\pm0.926}$ | $0.259_{\pm0.097}$ | $0.714_{\pm0.241}$ | $0.467_{\pm0.149}$ | $0.450_{\pm0.090}$ | $0.341_{\pm0.075}$ | $0.373_{\pm0.076}$ |
| $\Delta T = 10\,000$ | MACE-OFF | $143.413_{\pm3.313}$ | $142.569_{\pm5.584}$ | $136.380_{\pm3.964}$ | $140.288_{\pm9.136}$ | $140.103_{\pm3.398}$ | $140.372_{\pm4.754}$ | $141.598_{\pm1.020}$ |
| | AMBER | $133.900_{\pm3.815}$ | $128.385_{\pm3.812}$ | $121.095_{\pm0.968}$ | $121.607_{\pm1.195}$ | $120.319_{\pm1.591}$ | $120.800_{\pm0.398}$ | – |
| | Lennard-Jones | $7.797_{\pm0.369}$ | $5.533_{\pm0.425}$ | $5.167_{\pm0.221}$ | $7.453_{\pm0.376}$ | $6.721_{\pm0.418}$ | $6.477_{\pm0.204}$ | $5.921_{\pm0.522}$ |
| | ATOM | $2.554_{\pm0.945}$ | $1.002_{\pm0.184}$ | $1.634_{\pm0.200}$ | $1.098_{\pm0.209}$ | $1.116_{\pm0.083}$ | $0.964_{\pm0.185}$ | $0.990_{\pm0.171}$ |

Table 18: Seconds to produce timestep $\Delta T = 3000$ of each MD22 trajectory at `float32` precision.

| | Ac-Ala3-NHME | DHA | Stachyose |
|---|---|---|---|
| MACE-OFF | $44.737_{\pm4.615}$ | $42.042_{\pm0.720}$ | $47.913_{\pm0.546}$ |
| Lennard-Jones | $4.700_{\pm0.212}$ | $3.790_{\pm0.246}$ | $6.613_{\pm0.147}$ |
| ATOM | $1.302_{\pm0.766}$ | $0.914_{\pm0.055}$ | $3.075_{\pm0.080}$ |

### F.4 ATOM HYPERPARAMETERS

We employ the same dataset splitting and discretization parameters reported in Xu et al. (2024) for the MD17. We set the batch size to 192, use the AdamW-AMSGrad optimizer (Loshchilov & Hutter, 2017) with an $\epsilon$ of $1 \times 10^{-10}$ to avoid instability associated with the small gradients produced by zero-initialised weight matrices in early training (Jordan et al., 2025). During multitask training, we reduce the number of epochs to 250 and employ the Muon optimizer (Jordan et al., 2024; Kim, 2021). We present a complete overview of our hyperparameters in Table 19.

Table 19: Hyperparameters for ATOM. MD17 hyperparameters are shared across all molecules unless otherwise noted.

| Module | | MD17, RMD17, TG80 | TG80 Multitask |
|---|---|---|---|
| Training | | | |
| | Batch size | 192 | 192 |
| | Epochs | 1000 | 250 |
| | Max grad norm | 1.0 | 1.0 |
| | Label noise $\sigma$ | 0.1 | 0.1 |
| | $\Delta t$ | 3000 | 10 000 |
| | Timesteps $P$ | 8 | 8 |
| | Train/Val/Test | $(500, 3\,000, 3\,000)$ | $(6\,500, 13\,000, 13\,000)$ |
| | RWPE length | 8 | 8 |
| Optimiser | | | |
| | Optimiser type | AdamW-AMSGrad | Muon |
| | Learning rate | $1 \times 10^{-3}$ | $1 \times 10^{-3}$ |
| | $\beta_1, \beta_2$ | 0.9, 0.999 | $(0.9, 0.999)$ |
| | Weight decay | $1 \times 10^{-5}$ | $1 \times 10^{-5}$ |
| | $\epsilon$ | $1 \times 10^{-10}$ | $1 \times 10^{-5}$ |
| Model | | | |
| | Embedding dim | 128 | 256 |
| | No. layers | 5 | 6 |
| | No. attention heads | 8 | 8 |
| | No. output heads | 1 | 8 |
| | Attention dropout | 0.2 | 0.2 |
| | RoPE frequency | 1000 | 1000 |
| | MLP layers | 2 | 2 |
| | MLP activation | SwiGLU | SwiGLU |
| | MLP dropout | 0.0 | 0.0 |
| | Norm type | RMS norm | RMS norm |
| | Learnable value residuals | True | True |

## F.5 EGNO HYPERPARAMETERS AND EXPERIMENTAL DETAILS

We generated the EGNO results reported in Table 1 with the same discretization parameters and hyperparameters as used in their experiments. We reduce the number of epochs from 10 000 to 2 500, use a batch size of 192 with the AdamW-AMSGrad optimizer (Loshchilov & Hutter, 2017), and select the best validation loss epoch for testing. In the multitask case, we further reduce the number of epochs to 250 and employ the Muon optimizer (Jordan et al., 2024; Kim, 2021). Complete hyperparameters are displayed in Table 20.

Table 20: Hyperparameter values for EGNO across each benchmark dataset.

| Module | | MD17, RMD17, TG80 | TG80 Multitask |
|---|---|---|---|
| Training | | | |
| | Batch size | 192 | 192 |
| | Epochs | 2500 | 250 |
| | Max grad norm | Uncapped | Uncapped |
| | Label noise $\sigma$ | 0.1 | 0.1 |
| | $\Delta t$ | 3000 | 10 000 |
| | Timesteps $P$ | 8 | 8 |
| | Train/Val/Test | $(500, 3\,000, 3\,000)$ | $(6\,500, 13\,000, 13\,000)$ |
| | RWPE length | 8 | 8 |
| Optimiser | | | |
| | Optimiser type | AdamW-AMSGrad | Muon |
| | Learning rate | $1 \times 10^{-3}$ | $1 \times 10^{-3}$ |
| | $\beta_1, \beta_2$ | $(0.9, 0.999)$ | $(0.9, 0.999)$ |
| | Weight decay | $1 \times 10^{-5}$ | $1 \times 10^{-5}$ |
| | $\epsilon$ | $1 \times 10^{-10}$ | $1 \times 10^{-5}$ |
| Scheduler | | | |
| | Scheduler type | StepLR | StepLR |
| | Step size | 2500 | 2500 |
| | $\gamma$ | 0.5 | 0.5 |
| Model | | | |
| | Embedding dim | 64 | 64 |
| | No. EGNO layers | 5 | 5 |
| | Temporal convolution activation | LeakyRELU | LeakyRELU |
| | MLP layers | 2 | 2 |
| | MLP activation | SiLU | SiLU |
| | MLP dropout | 0 | 0 |
| | Time embedding dim | 32 | 32 |
| | Fourier modes | 2 | 2 |

# G PROPOSITIONS AND PROOFS

## G.1 KERNEL INTEGRAL FORM OF CROSS-ATTENTION

**Proposition G.1.** *The cross-attention is equivalent to a kernel integral operator, i.e.,* $\text{softmax}(\text{T-RoPE}(\mathbf{Q})\,\text{T-RoPE}(\mathbf{K}_i)^\top/\sqrt{d_h})\mathbf{V}_i = \int \kappa_i(\mathbf{z}, \mathbf{x})v_i(x)d\mu_N(\mathbf{x})$, *where* $\kappa_i$ *denotes the kernel induced by softmax function,* $v_i(\mathbf{x})$ *denotes the values as a function of* $\mathbf{x}$, *and* $\mu_N$ *denotes the empirical measure supported on* $\{\mathbf{x}_j\}_{j=1}^N$.

*Proof of Proposition G.1.* Following (Gao et al., 2024) we may view our attention as a kernel integral transform by considering $\mathbf{x}_i$ as being sampled from the continuum domain $\Omega \subset \mathbb{R}^3$ for which we define the empirical measure with support on $\{\mathbf{x}_j\}_{j=1}^N \subset \Omega$:

$$\mu_N(\mathbf{x}) = \frac{1}{N}\sum_{i=1}^N \delta_{\mathbf{x}_j}, \qquad \int_\Omega g(\mathbf{x})d\mu_N(\mathbf{x}) = \frac{1}{N}\sum_{j=1}^N g(\mathbf{x}_j) \tag{12}$$

where $\delta$ is the Dirac delta function "selecting" the values at $\mathbf{x}_j$. Given T-RoPE-rotated query and key maps $\tilde{q}_\theta(\mathbf{z}) = R_{p(\mathbf{z})}q_\theta(\mathbf{z}_j)$, $\tilde{k}_i(\mathbf{x}_j) = R_{p(\mathbf{x}_j)}k_{\theta,i}(\mathbf{x}_j)$ we form the data-dependent kernel for feature $F$:

$$\kappa_{\theta,i}(\mathbf{z}, \mathbf{x}_j) = \frac{\exp\big(\langle\tilde{q}(\mathbf{z}),\,\tilde{k}_i(\mathbf{x}_j)\rangle\,/\,\sqrt{d_h}\big)}{\displaystyle\int_\Omega \exp\big(\langle\tilde{q}(\mathbf{z}_j),\,\tilde{k}_i(\mathbf{x}')\rangle\,/\,\sqrt{d_h}\big)\,d\mu_N(\mathbf{x}')}\,. \tag{13}$$

Thus, for any $F \in \mathcal{F}$ we may represent our cross-attention as the kernel integral operator:

$$\big(\mathcal{K}_{\theta,i}\mathbf{v}_j\big)(\mathbf{z}) = \int_\Omega \kappa_{\theta,i}\big(\mathbf{z}, \mathbf{x}\big)\,\mathbf{v}_i(\mathbf{x})\,\mathrm{d}\mu_N(\mathbf{x}), \qquad \int_\Omega \kappa_{\theta,i}(\mathbf{z}, \mathbf{x})\,d\mu_N(\mathbf{x}) = 1, \tag{14}$$

which is row-stochastic under the measure in Equation (12). $\qquad\square$

We remark that the kernel fails to satisfy global Lipschitz continuity (Delattre et al., 2023), unlike FNO (Li et al., 2021), and certain generalization theorems fail as a result (Le & Dik, 2024).

# H TRAJECTORY SAMPLES

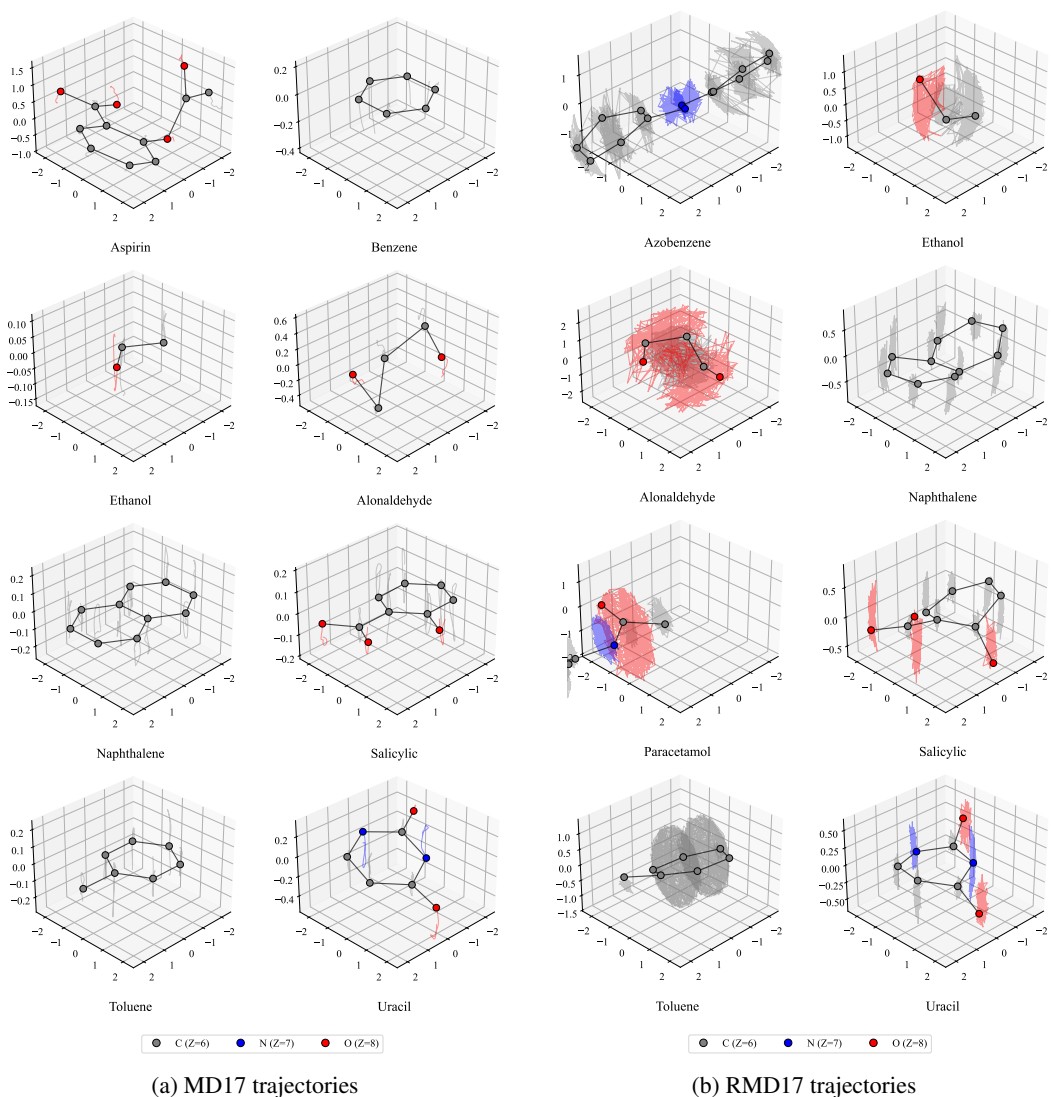

(a) MD17 trajectories                    (b) RMD17 trajectories

Figure 10: 3000 steps MD trajectories from the MD17 and RMD17 datasets.

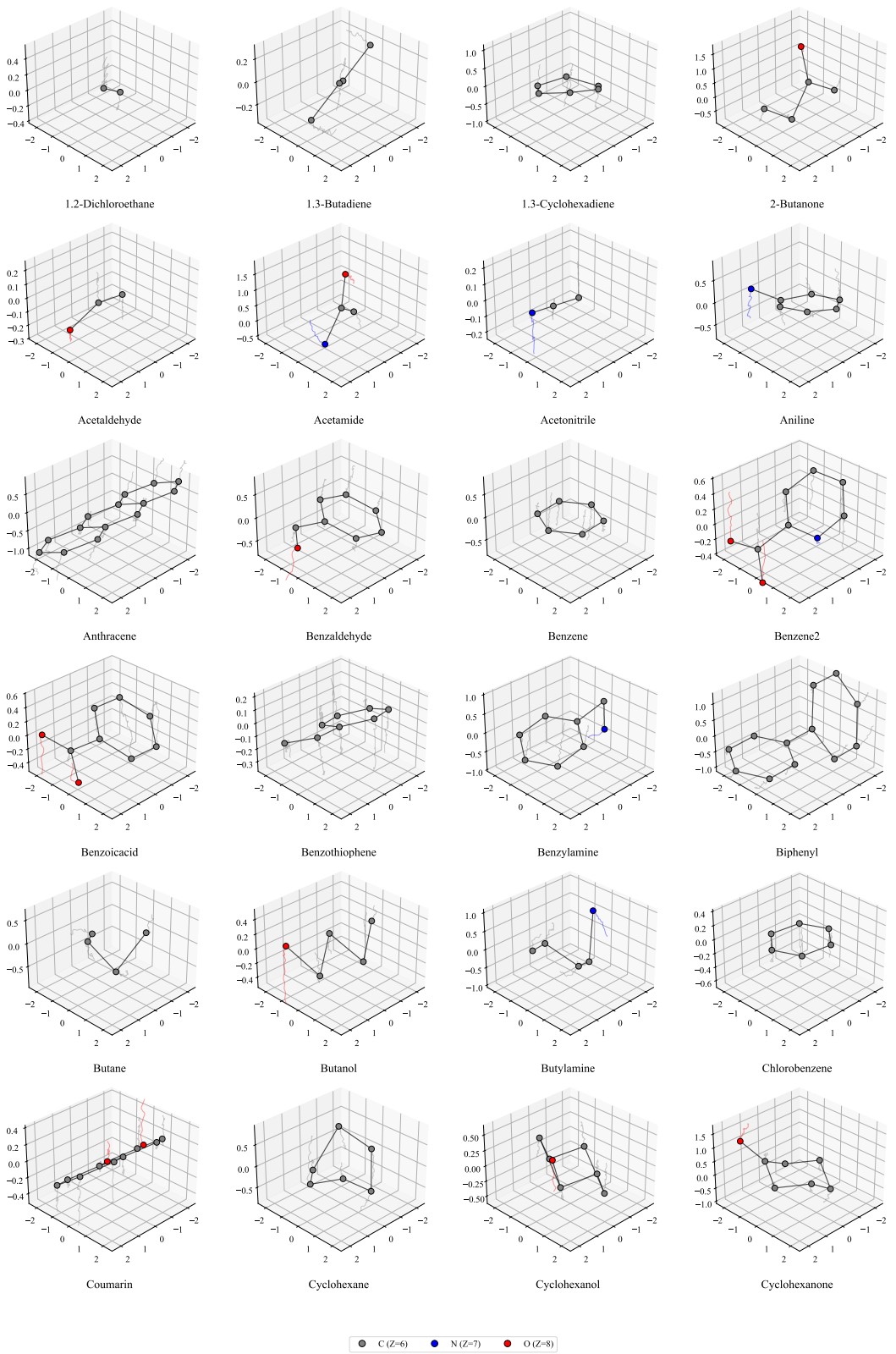

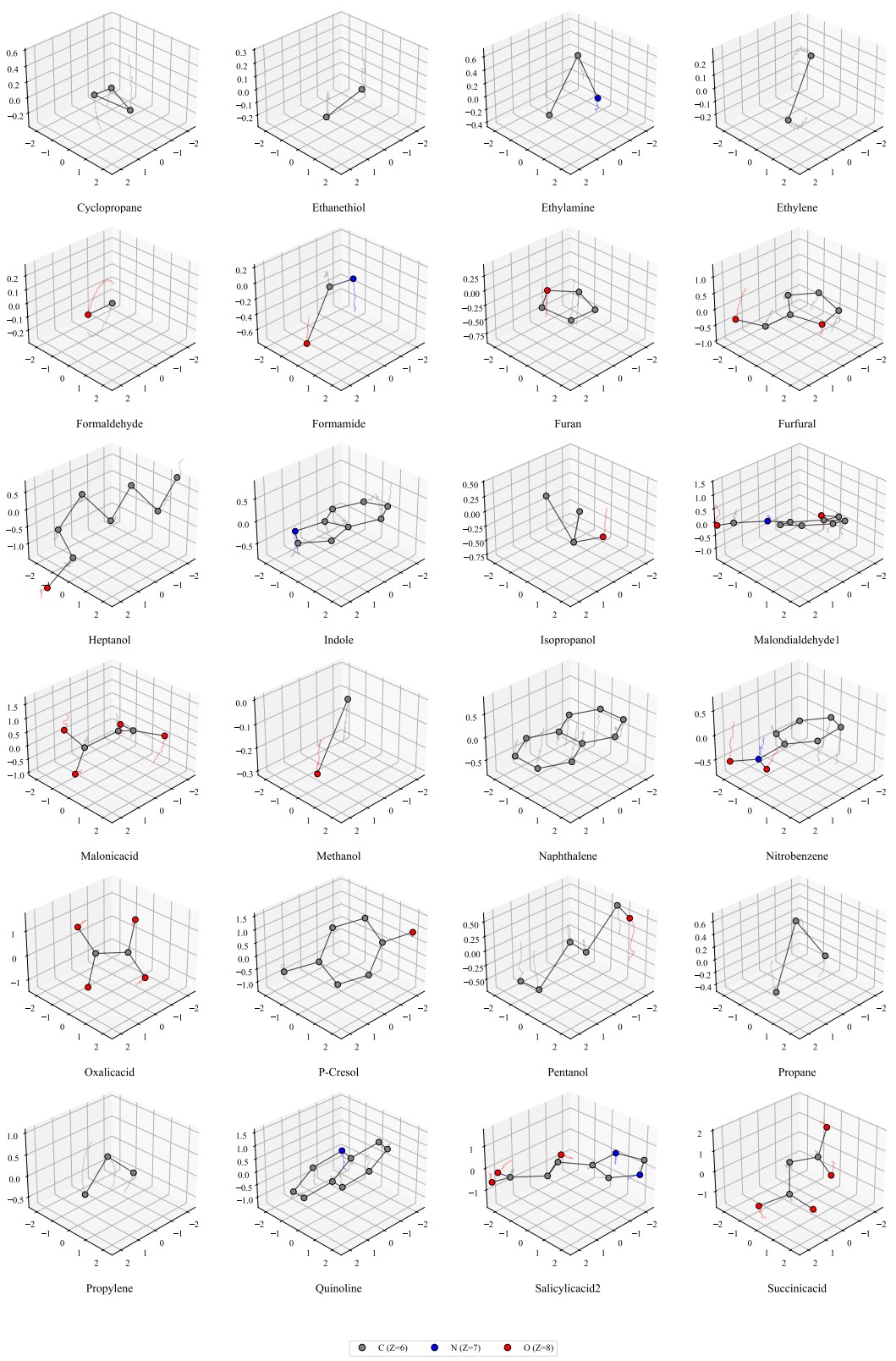

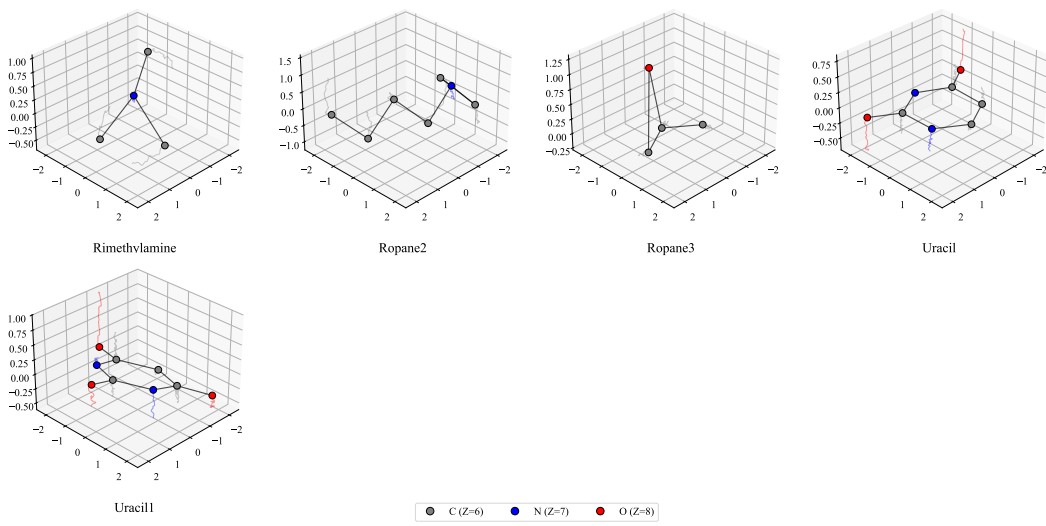

Figure 11: 3000-step MD trajectories from TG80. Molecules generated by our dataset expansion algorithm are named according to their seed molecule and the order of their selection.

