# OpenReview forum: "ATOM: A Pretrained Neural Operator for Multitask Molecular Dynamics"
_ICLR.cc/2026/Conference — ICLR 2026 Poster_

### Official Review · Reviewer_hg7v · 2025-10-15

**Soundness:** 2
**Presentation:** 2
**Contribution:** 2
**Rating:** 4
**Confidence:** 4

**Summary:**

This work presents the atomistic transformer operator for molecules (ATOM), which preserves quasi-equivariance and enables parallel decoding of molecule states across multiple timesteps. The model leverages E(3)-equivariant linear layers to produce symmetry-aware features, and proposes temporal RoPE (T-RoPE) to build the heterogeneous temporal attention mechanism. For both single-task experiments on MD17, MD22 and multitask experiments on the curated TG80 dataset, ATOM consistently outperforms the selected baselines, showing an out-of-distribution generalization across unseen molecules.

**Strengths:**

- The paper is written with a clear logical flow, facilitating reader comprehension. Each component of the model architecture is well justified and supported by comprehensive ablation studies.
- The proposed T-RoPE method ingeniously enables attention to achieve translation equivariance along the temporal dimension, thereby supporting parallel prediction of future molecular states at various timesteps.
- The newly introduced TG80 dataset represents a valuable extension over datasets such as MD17 and MD22 in terms of both scale and trajectory accuracy, contributing to the advancement of the field.
- ATOM demonstrates out-of-distribution generalization on the TG80 dataset, which is crucial for practical applications.

**Weaknesses:**

- The authors claim that ATOM enables parallel computation through the T-RoPE technique, thereby overcoming the limitation of sequential sampling. However, this claim lacks experimental evidence. For instance, it would be helpful to compare the inference time of ATOM with that of sequential generative models when generating trajectories of equal wall-clock duration.
- Given the molecular state at time $t$, the model outputs future states at different time intervals based on the assumption that the trajectory is uniquely determined by the initial state. However, in realistic scenarios like protein–ligand interactions in aqueous solution, thermal fluctuations must be taken into account, which are typically modeled using Langevin dynamics. In such cases, the trajectory is not uniquely determined but rather represented as a probabilistic distribution. Therefore, the training and evaluation of ATOM are not fully aligned with real physical settings.
- As the authors mentioned in the Limitations section, ATOM has not yet been validated on larger molecular systems. However, for small-molecule systems with fewer than 20 heavy atoms, MD simulations using empirical force fields are already highly efficient. The authors should therefore further clarify the advantages of ATOM over classical MD simulations based on empirical force fields.

**Questions:**

1. Whether ATOM offers efficiency improvements over sequential generative models remains to be verified. The authors are encouraged to provide a comparison of inference time between ATOM and conventional sequential generation models when generating trajectories of equal wall-clock duration.
2. Although ATOM is theoretically capable of predicting future states at different time intervals through T-RoPE, the maximum predictable time interval is pre-defined as $\Delta T$. Therefore, when generating a long trajectory, using ATOM appears equivalent to employing a sequential generative model with a fixed time step of $\Delta T$. Please provide a reasonable explanation for this or correct me if I understand wrong.
3. In line 316, the experimental setup reports $\Delta T=3000$. Is this value expressed relative to the frame spacing of trajectories in the training datasets? What wall-clock time does it correspond to?
4. To my knowledge, for molecular systems with fewer than 20 heavy atoms, MD simulations using empirical force fields can already achieve high efficiency. Please clarify the advantages of ATOM over classical MD methods, preferably supported by additional experimental results.
5. In Table 3, EGNO appears to exhibit a much smaller performance drop from in-distribution to out-of-distribution settings compared to ATOM. Please provide an explanation for this observation.
6. ATOM predicts future states at different time intervals given the current state, which typically relies on the assumption that the trajectory is uniquely determined by the initial state. However, in broader application scenarios, biomolecules generally interact in aqueous solution, where thermal fluctuations must be considered. Thus, trajectories should be represented as distributions rather than being uniquely determined. ATOM appears not directly applicable to such settings. How do the authors address this limitation?

**Details Of Ethics Concerns:**

None.

---

> ### Author Response · Authors · 2025-11-21
> **Response to Reviewer hg7v (Part I)**
>
> We thank the reviewer for the insightful comments and positive assessment of our paper’s clarity, architecture, use of T-RoPE for parallel decoding, the TG80 extension, and ATOM’s out-of-distribution generalization. Below, we provide detailed responses to your comments and questions.
>
>
> **1. (W1 + Q1) Experiment on the benefit of parallel decoding compared to sequential models**
>
> **Reply**: Thank you for the suggestion. We have now included an inference-time comparison of ATOM against autoregressive baselines: Lennard-Jones potentials [2] and MACE [1], a popular pretrained MLIP on MD17 and MD22. As shown in Table 18, ATOM is substantially faster than these autoregressive methods, achieving on average a **~20x speedup** over MACE and a **~5x speedup** over Lennard-Jones in wall-clock simulation time, highlighting the efficiency of ATOM’s fully parallel decoding. We intend to add additional force-field baselines (e.g., AMBER) in the final manuscript.
>
> [1] Kovács, et al. 2025. Mace-off: Short-range transferable machine learning force fields for organic molecules. Journal of the American Chemical Society, 147(21), 17598-17611.
>
> [2] Schwerdtfeger & Wales. 2024. 100 years of the Lennard-Jones potential. Journal of Chemical Theory and Computation, 20(9), 3379-3405.
>
> ---------------------
> **2. (W2 + Q6) On the cases where trajectories are distributions**
>
> **Reply**: ATOM is deliberately formulated and evaluated in the deterministic regime, as all TG80 trajectories are *ab initio* DFT simulations, where future states are **uniquely determined** by the initial state. In this setting, a deterministic neural operator that maps an initial segment of the trajectory to future states is *entirely consistent with the data-generating process*. This setting is standard in time-coarsened MD, including FlashMD, TrajCast, and other methods [1, 2]. We have added a section explicitly connecting our work to these approaches in the related work section.
>
> Nonetheless, we agree that many realistic systems are more naturally modeled by stochastic dynamics such as Langevin equations. Extending ATOM to this setting is, in principle, feasible by training on trajectories generated with a Langevin thermostat and adapting the operator to predict a distribution over future states. However, this would require re-simulating the entire TG80 dataset under stochastic dynamics, which is beyond our current resources or allowed time for the rebuttal period (TG80 generation takes ~6 weeks per molecule). We now discuss this limitation explicitly and highlight stochastic extensions of ATOM as an important direction for future work.
>
> [1] Bigi et al. 2025. “FlashMD: long-stride, universal prediction of molecular dynamics”. *NeurIPS 2025*.
>
> [2] Thiemann et al. 2025. “Force-Free Molecular Dynamics Through Autoregressive Equivariant Networks”. *arXiv:2503.23794*.
>
> ------------------
> **3. (W3 + Q4) Comparison to classical MD simulation for small-molecule system**
>
> **Reply:** We thank the reviewer for raising this critical point. In classical MD, the cost of generating a long trajectory scales linearly with the number of integration steps, and empirical force fields still require very small timesteps (typically 0.5–2 fs) for stability. In contrast, ATOM directly predicts long-stride trajectories (e.g., 3000 fs) in a *single* forward pass. In the revision (Tables 18 and 19), we have added inference-time experiments comparing classical force fields, a pretrained MLIP (MACE), and ATOM for producing 3000 fs trajectories (ATOM with $\Delta T = 3000$ fs). In both MD17 (small molecule) and MD22 (large molecule), ATOM yields significantly better wall-clock time, achieving an average of **~20x speedup** over MACE.
>
> ----------------------
> **4. (Q2) “ATOM appears equivalent to employing a sequential generative model with a fixed time step of $\Delta T$”**
>
> **Reply**: We do not believe ATOM is equivalent to sequentially calling a model with fixed $\Delta T$ as T-RoPE allows for an arbitrary choice of $\Delta T$ at inference time. We have updated Figure 3 to clarify this, showing the performance of a pretrained $\Delta T = 3000$ ATOM with $\Delta T$ up to 10 000 at inference. We highlight that ATOM accomplishes this with a single forward pass, whereas a sequential model would require at least four $\Delta T = 3000$ forward passes to reach 10 000.
>
> ----------------------------
> **5. (Q3) Clarification on setup with $\Delta T = 3000$**
>
> **Reply**:  $\Delta T$ is the gap between the initial frame and future frame to be predicted in the training data. To clarify the wallclock time this corresponds to, we have added femtosecond units to lines 316 and 358.

---

> > ### Author Response · Authors · 2025-11-21
> > **Response to Reviewer hg7v (Part II)**
> >
> > **6. (Q5) “EGNO appears to exhibit a much smaller performance drop from in-distribution to out-of-distribution settings compared to ATOM”**
> >
> > **Reply**:  We thank the reviewer for this observation. The smaller ID to OOD drop for EGNO in Table 3 should not be interpreted as better out-of-distribution robustness, but rather as a consequence of *EGNO substantially underfitting even in-distribution*.
> >
> > In the ID regime, ATOM improves over EGNO by 78–89% S2T MSE across all five clusters. In the OOD regime, ATOM still improves over EGNO by 22–59% (e.g., Cluster 3: 60.95 vs. 151.74), and *OOD ATOM even outperforms ID EGNO* in 4 out of 5 folds. Thus, ATOM’s larger relative drop reflects that it starts from a much lower ID error, not that it generalizes worse. In absolute terms, its OOD performance remains far superior to EGNO.

---

> > > ### Comment · Reviewer_hg7v · 2025-11-21
> > >
> > > Thanks for the authors' response. The explanations and additional experiments have indeed addressed some of my concerns. Below are several points on which I would still like to engage in further discussion:
> > > 1. In the reply to Q6, the authors claim that "as all TG80 trajectories are ab initio DFT simulations, where future states are uniquely determined by the initial state." This assertion is conceptually inaccurate. Ab initio DFT provides only the energies and forces for a given conformation, and whether a trajectory is deterministic or stochastic is determined by the dynamical equations employed. The determinism of the TG80 trajectories arises from vacuum simulations where no damping terms need to be considered, and is therefore unrelated to the use of DFT itself. While I acknowledge that ATOM's deterministic neural operator is consistent with the current experimental setting, realistic physical systems generally require incorporating solvent effects and the associated stochastic forces. This raises concerns regarding the practical applicability of ATOM beyond vacuum environments. Moreover, several publicly available datasets provide MD trajectories in explicit solvent, such as Alanine-dipeptide (https://markovmodel.github.io/mdshare/ALA2/) and ATLAS (https://www.dsimb.inserm.fr/ATLAS), which could be considered for future evaluation.
> > > 2. I find the authors' clarification in response to Q2 meaningful, particularly the point that the model can generate trajectories with a larger inference-time time lag $\Delta T$ than that used during pretraining. However, this notable property of the model is not clearly articulated in the manuscript. The current description in Section 4.3 is ambiguous and may mislead readers into believing that the comparison involves different pretraining $\Delta T$ values rather than different inference-time choices.
> > > 3. Regarding Tables 18 and 19, I appreciate the added comparison of inference time between ATOM, the machine-learning force field (MACE), and the classical force field (LJ potential). Nonetheless, this comparison alone is insufficient. Higher inference efficiency does not necessarily translate into practical utility unless accompanied by comparable predictive accuracy, and the manuscript does not establish that ATOM achieves accuracy on par with MACE or the LJ potential. To more convincingly demonstrate the model's effectiveness, I recommend including evaluations of the similarity between ATOM-generated trajectories and those obtained via standard force fields (e.g., AMBER) in MD simulations.
> > >
> > > In summary, I will maintain my score for now, and I look forward to further discussions with the authors.

---

> ### Author Response · Authors · 2025-11-23
> **Further Response to Reviewer hg7v (Part I)**
>
> We are glad our responses have addressed some of your concerns. We have taken the time to run several further experiments to ensure adequate comparison with current practical methods. Regarding your further comments, we would like to address them below.
>
> **1.** Thank you for this insightful comment. We fully acknowledge that stochastic (e.g., Langevin) formulations are standard in many practical ab initio MD applications. In this work, however, ATOM is positioned within the line of **deterministic time-coarsening methods**, and our framework is consistent with this setting, as adopted in recent works such as [1,2,3,4,5,6].
>
> The datasets highlighted by the reviewer (e.g., alanine dipeptide and ATLAS) primarily contain protein or peptide trajectories in explicit solvent, whereas TG80 focuses on long-timespan dynamics (32 000 fs) of small drug-like and solvent molecules (e.g., aspirin, nitrobenzene). This places our work in the lineage of MD17, RMD17, and MD22, rather than ATLAS and similar datasets, which focus on protein dynamics. We now clarify this distinction in the paper and *explicitly list extensions of ATOM to explicit-solvent, stochastic systems as an important direction for future work*.
>
> Specifically, as ATOM is a propagation operator over positions, it does not directly output forces, so it cannot be plugged into a standard Langevin integrator in ASE out of the box. In the Limitations, we propose adding a framewise energy head $E_\theta(x_t)$, with forces obtained via
> $$
> F_t^{(p)} = - \nabla_{x_t^{(p)}} E_\theta(x_t),
> $$
> for each atom $p$. This yields exactly the quantities ASE expects from a calculator (potential energy and forces), so ATOM+$E_\theta$ can be wrapped as an ASE Calculator and used with ASE’s Langevin MD routines. The corresponding underdamped Langevin dynamics are
> $$
> \mathrm{d} x_t^{(p)} = v_t^{(p)} \mathrm{d} t,\qquad
> m_p \mathrm{d} v_t^{(p)} =
> F_t^{(p)} \mathrm{d} t
> -\gamma_p  m_p  v_t^{(p)} \mathrm{d} t
> +\sqrt{2 \gamma_p  m_p  k_B T}  \mathrm{d} W_t^{(p)},
> $$
> where $x_t^{(p)}$ and $v_t^{(p)}$ are position and velocity, $m_p$ is mass, $\gamma_p$ friction, $k_B$ Boltzmann’s constant, $T$ temperature, and $W_t^{(p)}$ is a standard Brownian. We believe this formulation provides a concrete formulation to apply ATOM to stochastic dynamics, and to integrate it into a widely used MD framework (ASE).
>
>
> --------------------------------------
> **2.** We are delighted to see that you find our clarifications meaningful, and thank you for your suggestion. We have now explicitly commented in Section 4.3 that ATOM is pretrained with a fixed $\Delta T = 3000$ and evaluated with varying $\Delta T$ **at inference**. We have also noted this feature explicitly in the comparison to EGNO section, on lines 284-287.
>
> ------------------------------------------
> **3.** We fully agree that runtime is only meaningful when considered together with accuracy. We now report MACE [7] accuracy on MD17 single-task and TG80 multitask (included in Table 1 and Table 3, highlighted in red).
> * On MD17 single-task with $\Delta T = 3000$ fs (Table 1), ATOM attains **comparable MSEs to MACE**, while being orders of magnitude faster at inference.
> * On TG80 long-time OOD multitask with $\Delta T = 10 000$ fs (Table 3), ATOM roughly **halves the MSE of MACE**. We attribute this larger gap in OOD to the increased error accumulation at $10\ 000$ fs versus the $3\ 000$ fs of MD17.
>
> Standard empirical force fields (LJ, AMBER/GAFF) target different potential energy surfaces from our vacuum DFT ground truth [8]. In our tests, their rollouts *diverge rapidly*, yielding very large MSEs that primarily reflect this physics mismatch. Accordingly, we retain LJ/AMBER timings only as practical latency references and focus accuracy comparisons on DFT-trained MLIPs such as MACE. In the final version, we will additionally include NequIP and other MLIP baselines, as well as error-accumulation plots, to provide a more comprehensive comparison.

---

> > ### Author Response · Authors · 2025-11-24
> > **Further Response to Reviewer hg7v (Part II)**
> >
> > References
> >
> > [1] Bigi, Filippo, Sanggyu Chong, Agustinus Kristiadi, and Michele Ceriotti. 2025. “FlashMD: Long-Stride, Universal Prediction of Molecular Dynamics”. arXiv preprint arXiv:2505.19350.
> >
> > [2] Liu, Yang, Jiashun Cheng, Haihong Zhao, et al. 2024. “SEGNO: Generalizing Equivariant Graph Neural Networks with Physical Inductive Biases”. arXiv preprint arXiv:2308.13212.
> >
> > [3] Thiemann, Fabian L., Thiago Reschützegger, Massimiliano Esposito, Tseden Taddese, Juan D. Olarte-Plata, and Fausto Martelli. 2025. “Force-Free Molecular Dynamics Through Autoregressive Equivariant Networks”. arXiv preprint arXiv:2503.23794.
> >
> > [4] Tuckerman, M., B. J. Berne, and G. J. Martyna. 1992. “Reversible Multiple Time Scale Molecular Dynamics”. The Journal of Chemical Physics.
> >
> > [5] Xu, Minkai, Jiaqi Han, Aaron Lou, et al. 2024. “Equivariant Graph Neural Operator for Modeling 3D Dynamics”. arXiv preprint arXiv:2401.11037.
> >
> > [6] Zheng, Tianze, Weihao Gao, and Chong Wang. 2021. “Learning Large-Time-Step Molecular Dynamics with Graph Neural Networks”. arXiv preprint arXiv:2111.15176.
> >
> > [7] Batatia, Ilyes, Dávid Péter Kovács, Gregor N. C. Simm, Christoph Ortner, and Gábor Csányi. 2023. “MACE: Higher Order Equivariant Message Passing Neural Networks for Fast and Accurate Force Fields”. arXiv preprint arXiv:2206.07697.
> >
> > [8] Chmiela, et al. 2018. “Towards exact molecular dynamics simulations with machine-learned force fields”. Nature Communications.

---

> > > ### Comment · Reviewer_hg7v · 2025-11-24
> > >
> > > Thank you for the prompt response. I believe the additional explanations and experiments have satisfactorily addressed my concerns. It is somewhat unfortunate that the application of ATOM to stochastic dynamics remains largely theoretical at this stage. Nonetheless, the work presented is already fairly comprehensive, and I will raise my score accordingly.

---

### Official Review · Reviewer_L5rw · 2025-10-30

**Soundness:** 3
**Presentation:** 2
**Contribution:** 3
**Rating:** 4
**Confidence:** 3

**Summary:**

The paper introduces ATOM, a pretrained neural operator designed to improve molecular dynamics (MD) modeling by incorporating approximate geometric equivariance. The method proposes a heterogeneous temporal attention mechanism along with a temporal RoPE (T-RoPE) adapation, and further enables large-scale pretraining and generalization across different MD tasks. Experimental results demonstrate that ATOM achieves competitive performance on multiple benchmarks compared to strong baselines such as EGNO.

**Strengths:**

* The paper addresses a critical problem, that is, improving neural operators for molecular dynamics, and further enables pretraining on large-scale datasets, which is an important step toward generalizable molecular representation learning.

* This paper constructs a new large-scale molecular dynamics dataset, TG80, which not only facilitates the pretraining of ATOM itself but also provides valuable resources for future research in this area.

* The empirical results are generally strong and show that ATOM performs well across multiple benchmarks, indicating the effectiveness of the proposed approach.

**Weaknesses:**

* **Unclear Architecture Design.** The paper repeatedly employs the term **quasi-equivariant** without providing a clear mathematical definition. Moreover, while the ablation studies compare various architectural variants, the main text and appendix lack a detailed, mathematically grounded description of the original model design. The absence of released code further limits reproducibility. The authors should clearly define quasi-equivariance, explain its advantages, and elaborate on how it is concretely maintained in the model.

* **Limited Discussion of Related Works.** The problem studied in the paper can also be framed as a **Time-Coarsened Dynamics** problem, yet many relevant works in this area are not sufficiently discussed [A, B, C]. The authors are encouraged to analyze the connections and differences between ATOM and prior works, including distinctions in problem formulation, modeling strategy, and evaluation metrics.

[A] Klein, Leon, et al. "Timewarp: Transferable acceleration of molecular dynamics by learning time-coarsened dynamics." Advances in Neural Information Processing Systems 36 (2023): 52863-52883.
[B] Hsu, Tim, et al. "Score dynamics: Scaling molecular dynamics with picoseconds time steps via conditional diffusion model." Journal of Chemical Theory and Computation 20.6 (2024): 2335-2348.
[C] Yu, Ziyang, Wenbing Huang, and Yang Liu. "UniSim: A Unified Simulator for Time-Coarsened Dynamics of Biomolecules." Forty-second International Conference on Machine Learning.

**Questions:**

*  In the single-task setting, what is the key difference between ATOM and the main baseline EGNO? Can ATOM degenerate into EGNO if some critical components are removed? If so, such a description would clarify the key improvements and help highlight the method’s contribution.

* Does multi-task training improve the performance of each individual task? The paper does not include an ablation study that isolates the effect of multi-task learning itself.

---

> ### Author Response · Authors · 2025-11-21
> **Response to Reviewer L5rw (Part I)**
>
> We thank the reviewer for reviewing our paper, for recognizing our work as “addressing a critical problem,” and for finding the TG80 dataset valuable and the empirical results strong. Here are our detailed responses to your comments.
>
> ------------------
> **1. Unclear architecture design**
>
> **Reply**: Thank you for raising this point. We have revised the paper to give a precise definition of quasi-equivariance, a mathematically grounded description of ATOM, and to clarify reproducibility.
>
> * **Quasi-equivariance**: In Section 3.1, we have now defined $\varepsilon$-quasi-equivariance as a bound on the expected equivariance error of a map $f$  as
> $$\mathbb E_{x \in \mathcal{X}} \|  \int_G f( \phi(g)(x) ) d \mu (g)  - \int_G \rho(g) (f(x)) d\mu(g) \| \leq \varepsilon$$
> where $\mu$ is the Haar measure on $G$. The case $\varepsilon = 0$ recovers standard equivariance. This is motivated by the relaxed equivariance formulation used in recent work [1]. In Appendix E.3, we describe how we estimate $\varepsilon$ via Monte Carlo sampling over group elements and data points, and we report that quasi-equivariant ATOM (with equivariant lifting) has a $\varepsilon$ roughly 90% smaller than ATOM without equivariant lifting. This demonstrates that our equivariant lifting layers achieve near-equivariance, whilst leaving the following transformer blocks flexible, as evidenced in our ablation.
>
> [1] Elhag et al. 2025. “Relaxed Equivariance via Multitask Learning”. *arXiv:2410.17878*.
>
> * **Mathematical description of model design**: We have added a mathematical definition of ATOM to section 3.1, defining it as a composition of operators
> $$F_\theta \coloneqq \mathcal{P} \circ \sigma(\mathcal{K}_L) \circ \cdots \circ \sigma(\mathcal{K}_1) \circ \mathcal{Q}$$
> Where $\mathcal{Q}$, $\mathcal{P}$ denote equivariant lifting and projection, respectively, and $\mathcal{K}$ is the attention kernel we define in Appendix G.
>
> * **Code**: We *have already included the code in the supplementary material* for reproducibility. The dataset and all code for new experiments conducted during the rebuttal period will be made public upon acceptance of the paper.
>
> ---------------------
> **2. Limited discussion of related works**
>
> **Reply**: We thank the reviewer for their suggestions and have included a new section discussing these works in our related works section, titled “Time-coarsened Molecular Dynamics”. The added section covers the reviewer’s suggested references [A], [B], [C], as well as additional relevant references [1], [2], [3]. In summary, most relevant methods require the direct learning of forces and are inherently sequential. In contrast, ATOM can be interpreted as a *force-free* deterministic coarse-graining approach, wherein temporal pushforward is approximated by a learned propagation operator which is decoded *in parallel*.
>
> [1] Bigi et al. 2025. “FlashMD: long-stride, universal prediction of molecular dynamics”. *NeurIPS 2025*.
>
> [2] Thiemann et al. 2025. “Force-Free Molecular Dynamics Through Autoregressive Equivariant Networks”. *arXiv:2503.23794*.
>
> [3] Zheng et al. 2021. “Learning Large-Time-Step Molecular Dynamics with Graph Neural Networks”. *arXiv:2111.15176*.

---

> > ### Author Response · Authors · 2025-11-21
> > **Response to Reviewer L5rw (Part II)**
> >
> > **3. Key difference of ATOM to EGNO in a single-task setting**
> >
> > **Reply**: We thank the reviewer for the question. Although both methods are neural operators for 3D dynamics, their core architectural choices and inductive biases differ fundamentally.
> >
> > * **Spatial backbone (graph EGNN vs point-cloud transformer)**: EGNO is built by stacking EGNN layers with an equivariant Fourier temporal convolution, operating on an explicit molecular graph with fixed bond-type edges. In contrast, ATOM leverages fully connected attention over the point cloud, without relying on an explicit bond graph or edge labels. This difference in representation (graph vs. globally connected point cloud) is precisely what allows ATOM to mitigate oversquashing and capture long-range non-bonded interactions (now further evidenced in Table 2).
> >
> > * **Equivariance vs quasi-equivariance**: EGNO maintains strict SO(3)/E(3) equivariance throughout the network via EGNN layers and equivariant temporal kernels. ATOM instead uses a quasi-equivariant design: we enforce equivariance only in the initial lifting layer and then relax it in subsequent transformer blocks. This relaxed approach is not present in EGNO and is a key conceptual contribution of ATOM.
> >
> > * **Temporal modeling (Fourier temporal convolution vs T-RoPE attention)**: EGNO models time via a Fourier temporal convolution in the frequency domain, then combines this with EGNN message passing. ATOM instead uses Temporal RoPE (T-RoPE) in the attention module, which encodes time gaps directly as phase shifts and yields translation-invariant attention across irregular time lags.
> >
> > Thus, ATOM does not simply “degenerate” to EGNO by turning off a few modules. Our single-task ablations *verify the contribution of each component*, and our MD22 and TG80 results show that these design choices translate into stronger performance, especially for large, sparsely connected molecules and out-of-domain generalization across chemical space.
> > We have now included the above discussions at the end of Section 3.2, explicitly contrasting ATOM with EGNO.
> >
> > ----------------------------
> > **4. On the benefit of multitask training compared to single-task training**
> >
> > **Reply**: We thank the reviewer for raising this comment. Our goal with multitask ATOM is to make the model *practical* in the same sense as modern machine-learning interatomic potentials (MLIPs): with a fixed training budget, a single model covers many molecules and time horizons, rather than requiring a separate single-task model per system.
> >
> > In the revision, we have added an explicit ablation that isolates the effect of multitask learning by training  (i) a shared multitask ATOM (ID and OOD clusters) (ii) separate single-task ATOM models with the *same architecture and total compute*.
> >
> > The results in Appendix E.2 show that **multitask ATOM (with zero-shot inference) matches or slightly improves the per-task S2T loss** on Formic Acid. Thus, multitask training does not trade off accuracy for convenience: it preserves single-task performance while yielding a single, universal operator that is far more data- and compute-efficient and also enables zero-shot generalization to unseen molecules. In the final version, we plan to include additional experiments to verify this effect on more molecules.

---

> > > ### Comment · Reviewer_L5rw · 2025-11-26
> > >
> > > Thanks for the detailed responses, which addressed most of my concerns. Accordingly, I will raise my rating to 6.

---

### Official Review · Reviewer_h8zv · 2025-11-02

**Soundness:** 2
**Presentation:** 1
**Contribution:** 2
**Rating:** 6
**Confidence:** 2

**Summary:**

This paper introduces ATOM, a neural operator for molecular dynamics that introduces a quasi-equivariant architecture for better flexibility and simulation efficiency. Molecular systems are modeled as fully connected point clouds. It employs a equivariant lifting layer to embed atomic positions and velocities into a symmetry-aware latent space and a temporal attention mechanism to enable parallel decoding of multiple future states. ATOM demonstrates better performance than state-of-the-art approaches on standard MD benchmarks and shows that it can transfer to unseen molecules better than prior approaches. The authors also present TG80, a large-scale benchmark dataset with simulation data for a diverse set of molecules.

**Strengths:**

* This is an important and timely problem

* For a multitask set up, the authors demonstrate better performance that state-of-art baselines

* Parallel decoding can be more efficient

* An MD dataset, TG80, is produced, which intended to be chemically diverse and numerically stable to help in multitask pretraining and benchmarking tasks

**Weaknesses:**

* The choice and implications of modeling atoms as a point cloud is not discussed

* As a non-expert, I found few insightful discussions or learnings in the paper. It would be good if the authors discussed:
(1) the explicit novelty of each part of their architecture in comparison to prior works; (2) the importance of generality as opposed to specialized models; (3) the insights behind structure of the attention-based mechanism.

* It is not clear what "heterogeneous attention" means and its importance.

* There is no evaluation of the efficiency benefits of the proposed mechanism

* Figure 2 does not add much value to the paper

**Questions:**

Please address comments in weakness.

---

> ### Author Response · Authors · 2025-11-21
> **Response to Reviewer h8zv (Part I)**
>
> We thank the reviewer for your time reviewing this paper and for recognizing our work as “important and timely”. Below are our responses to your comments.
>
> -------------------------
> **1. The choice and implications of modeling atoms as a point cloud is not discussed**
>
> **Reply**: We chose to model molecules as point clouds (rather than a graph based on bond connectivity) to improve *information propagation*, especially in sparse molecular graphs with low bond connectivity (e.g., Figure 2, DHA), where long-range interactions are otherwise difficult to capture and oversquashing is severe. To substantiate this design choice, we have added MD22 ablations (in Table 2 of the revised manuscript), where we replace our point-cloud full attention with standard edge-based attention (GATv2). Across the benchmark, this substitution markedly degrades the performance of ATOM on sparsely connected molecules. These results support our claim that point-cloud modeling is crucial for alleviating oversquashing and capturing long-range effects in such regimes.
>
> ---------------------
> **2. More insightful discussions**
>
> **Reply**: We appreciate the reviewer for the suggestions to include more insightful discussions.
>
> (1) **Explicit novelty compared to prior works**: We have now included a comparison of the proposed ATOM to prior EGNO at the end of Section 3.2. Specifically, we highlight three main novelties over EGNO.
> * *First*, EGNO is a graph EGNN operating on fixed bond connectivity, whereas ATOM uses an E(3)-equivariant lifting layer followed by globally connected point-cloud attention, which better handles long-range and sparsely bonded interactions.
> * *Second*, EGNO is strictly equivariant end-to-end, while ATOM is quasi-equivariant, enforcing equivariance only in the lifting stage and relaxing it in deeper transformer layers, which our ablations show improves optimization and accuracy.
> * *Third*, EGNO models time via Fourier temporal convolution, whereas ATOM uses Temporal RoPE inside attention, giving translation-invariant handling of irregular time gaps and stronger temporal extrapolation. This also allows inference-time choice of time-horizon ($\Delta T$).  We also refer to our Reply #3 to Reviewer L5rw for further discussion.
>
> (2) **Importance of generality as opposed to specialized models**: Our goal with multitask ATOM is to make the model *practical* in the same sense as modern machine-learning interatomic potentials (MLIPs): with a fixed training budget, a single model covers many molecules and time horizons, rather than requiring a separate single-task model per system.
>
> In the revision, we have added an explicit ablation that isolates the effect of multitask learning by training (i) a shared multitask ATOM (ID and OOD clusters) (ii) separate single-task ATOM models with the *same architecture and total compute*. The results in Appendix E.2 show that multitask ATOM (with zero-shot inference) matches or slightly improves the per-task S2T loss on Formic Acid. Thus, multitask training does not trade off accuracy for convenience: it preserves single-task performance while yielding a single, universal operator that is far more data- and compute-efficient and also enables zero-shot generalization to unseen molecules. In the final version, we plan to include additional experiments to verify that this effect holds for further molecules.
>
>
> (3) **Insights behind the attention-based mechanism**: the design of our attention module is guided by three main insights.
> *  (i) *Capturing long-range interactions*. ATOM uses globally connected point-cloud attention, which mitigates over-squashing and captures non-bonded interactions (Also refer to Reply#1 and Reply#2(1)).
> * (ii) *Attention as a flexible neural operator*. Our architecture follows the recent work showing that transformer-based neural operators provide more expressive kernels than Fourier Neural Operators (FNO) in complex domains and time-dependent settings [1].
> * (iii) *Heterogeneous attention*. Attention makes it easy for the model to integrate geometric and dynamical information and to learn force-like interactions between atoms. Our ablations (Figure 5) show that replacing this heterogeneous attention with standard self-attention harms performance.
>
> [1] Hao et al. 2023. GNOT: A General Neural Operator Transformer for Operator Learning. *ICML 2023*.

---

> ### Author Response · Authors · 2025-11-21
> **Response to Reviewer h8zv (Part II)**
>
> **3. Explanation of “heterogeneous attention”**
>
> **Reply**: We appreciate the opportunity to clarify this point. By heterogeneous attention, we mean that ATOM jointly attends across **multiple channel types** (positions, velocities, and phase features), rather than treating everything as a single feature vector.
>
> Concretely, each channel type has its **own projection into query/key/value spaces**, and the attention module learns how to weight and mix geometric (position/velocity) and phase information when predicting future states. This differs from standard self-attention, which applies a single shared projection to a concatenated feature vector. In our ablation study (Figure 6), replacing heterogeneous attention with standard self-attention results in a performance degradation, indicating that explicitly modeling the heterogeneity of these channels is important for capturing the coupled structure of MD trajectories.
>
> ---------------------
> **4. Evaluation of efficiency**
>
> **Reply**: Thank you for the suggestion. We have now included an inference-time comparison of ATOM against two classic autoregressive baselines: Lennard-Jones potentials [2] and MACE [1], a popular pretrained MLIP, on MD17 and MD22. As shown in Tables 18 and 19, ATOM is substantially faster than these autoregressive methods, achieving on average a **~20× speedup** over MACE and a **~5× speedup** over Lennard-Jones and AMBER potentials in wall-clock simulation time, highlighting the efficiency of ATOM’s fully parallel decoding. We have additionally included loss comparisons with MACE, showing that this efficiency improvement **does not come at the cost of substantially reduced accuracy** on MD17 single-task or TG80 multitask learning.
>
>
> [1] Kovács, et al. 2025. Mace-off: Short-range transferable machine learning force fields for organic molecules. Journal of the American Chemical Society, 147(21), 17598-17611.
>
> [2] Schwerdtfeger & Wales. 2024. 100 years of the Lennard-Jones potential. Journal of Chemical Theory and Computation, 20(9), 3379-3405.
>
> --------------------
> **5. On Figure 2**
>
> **Reply**: Thank you for your suggestion. We have now moved Figure 2 to the Appendix.

---

### Author Response · Authors · 2025-11-21
**Summary of Changes**

We sincerely thank all reviewers for their constructive feedback. We have revised the manuscript accordingly, and all changes are highlighted in red. Below, we summarise the main modifications:

* As per suggestions by Reviewer L5rw and hg7v, we have expanded the **related work** section (Section 2) to discuss **time-coarsened molecular dynamics** and to more clearly position ATOM within this line of work.
* Following Reviewer L5rw’s comments, we now provide a **formal definition of quasi-equivariance** in terms of the expected equivariance error (Definition 3.1), describe a Monte Carlo estimator of this error in Appendix E.3, and **formally define ATOM as a composition of operators** in Section 3.2.
* In response to Reviewer h8zv and L5rw, we added an explicit **comparison to EGNO** in Section 3.2, highlighting key differences and novelties such as point-cloud heterogeneous attention, the quasi-equivariant design, and temporal RoPE.
* To answer the questions on **point-cloud full attention** (raised by Reviewer h8zv), we added an ablation comparing ATOM with full point-cloud attention to a variant using graph-based attention. Table 2 shows that graph-based attention significantly degrades performance, especially on sparsely connected molecules.
* Based on the comment by Reviewer hg7v, we have expanded the discussion on how ATOM could be extended to model **stochastic dynamics**.
* Based on the suggestions by Reviewer h8zv and hg7v, we added an **inference-time comparison** between ATOM and classical autoregressive baselines in Appendix F.3. ATOM is substantially faster than these baselines, due to its fully parallel decoding.
* In response to Reviewer h8zv and L5rw, we added an experiment in Appendix E.2 showing that **multitask ATOM** achieves similar or better performance than single-task ATOM, while being more data- and compute-efficient.
* Based on the suggestion by Reviewer hg7v, we added a **performance comparison to MACE** (a classic autoregressive baseline) for both ID and OOD settings (Table 1 and Table 3).

---

### Meta-Review · Area_Chair_omQt · 2026-01-02

**Summary:**

This paper proposes a new method named Atomistic Transformer Operator for Molecules (ATOM), which is a pretrained transformer neural operator for multi-task molecular dynamics. The reviewer raised some concerns about efficiency and accuracy comparison, mathematical details and stochastic dynamics. The authors provided a strong rebuttal to address these concerns and two reviewers believe the issues are addressed. Therefore, I tend to accept this paper.

**Reviewer Concerns:**

See above.

**Reviewer Scores:**

Two reviewers would increase their scores.

---

### Decision · Program_Chairs · 2026-01-26

Accept (Poster)